# Adaptive Momentum and Nonlinear Damping for Neural Network Training

**Aikaterini Karoni** [1]   **Rajit Rajpal** [2]   **Benedict Leimkuhler** [2]   **Gabriel Stoltz** [3]

## Abstract

Momentum Stochastic Gradient Descent (mSGD) relies on a fixed momentum coefficient shared across all parameters, failing to account for the heterogeneous structure of modern loss landscapes. In this work, we adopt a continuous-time formulation to introduce individual, adaptive momentum coefficients regulated by the kinetic energy of each model parameter. This mechanism automatically adjusts to evolving training dynamics to maintain stability without sacrificing convergence speed. We demonstrate that this adaptive friction is inextricably linked to cubic damping, a suppression mechanism from structural dynamics. We additionally introduce two optimization schemes by augmenting the continuous dynamics of mSGD and Adam with a cubic damping term. Empirically, our methods demonstrate robustness and match or outperform Adam on training ViT, BERT, and GPT2 tasks where mSGD typically struggles. We further provide theoretical results establishing the exponential convergence of the proposed schemes.

## 1. Introduction

Neural network loss landscapes are highly non-convex, multi-modal and difficult to navigate (Choromanska et al., 2015; Li et al., 2018). Improvements in training dynamics directly impact convergence speed, generalization performance, and computational cost. Momentum-based stochastic gradient methods are widely used to address these challenges. By maintaining exponentially weighted averages of past gradients, momentum stochastic gradient descent (mSGD) can accelerate progress along low-curvature directions and help control oscillations caused by stochastic

gradients and ill-conditioning (Sutskever et al., 2013). Recent work also shows that momentum can reduce stochastic-gradient bias (Shaw & Whalley, 2025). In practice, this allows for larger and more robust learning rate choices than standard SGD. However, if not carefully tuned, the inertia introduced through the momentum mechanism can itself become a source of oscillations in directions of high curvature or lead to excessive damping and slow convergence.

The standard practice in deep learning is to keep the momentum coefficient fixed during training, despite the presence of highly coordinate-dependent curvature. In this work, we argue that the momentum coefficient should not remain constant during training. Instead, it should be dynamically adapted to reflect the evolving kinetic energy of each parameter. Importantly, this adaptation should happen on a per-parameter basis to account for heterogeneity in the loss landscape.

The persistent performance gap between mSGD and Adam in transformer training has motivated several recent works seeking to explain this discrepancy (Zhang et al., 2024; Kunstner et al., 2023; Tomihari & Sato, 2025; Zhang et al., 2020). While one proposed explanation is that Adam is more robust to heavy-tailed gradient noise (Zhang et al., 2020), recent work (Kunstner et al., 2023) demonstrates that Adam's advantage persists even in the full gradient setting, where it behaves similarly to sign descent with momentum. This suggests that Adam's strength lies in its coordinate-wise normalization, which allows it to tackle the Hessian heterogeneity and anisotropic curvature that often cause mSGD to struggle (Zhang et al., 2020). As illustrated in Figure 1, our proposed methods substantially reduce this gap on the transformer tasks considered, without requiring explicit per-parameter adaptive learning rates.

To clarify the mechanics of our approach, we first establish the connection between momentum and friction (or damping). We begin with the discrete dynamics of mSGD (Poljak, 1964), which can be written as [1]

$$p_{n+1} = \mu p_n + \nabla f(x_n), \tag{1a}$$
$$x_{n+1} = x_n - \delta t\, p_{n+1}. \tag{1b}$$

In the zero–learning rate limit, these equations correspond to

[1]School of Mathematics, University of Bristol, Bristol, United Kingdom [2]School of Mathematics, University of Edinburgh, Edinburgh, United Kingdom [3]CERMICS, CNRS, École des Ponts, Institut Polytechnique de Paris; Inria Paris, France. Correspondence to: Katerina Karoni <ve24580@bristol.ac.uk>.

*Proceedings of the 43rd International Conference on Machine Learning*, Seoul, South Korea. PMLR 306, 2026. Copyright 2026 by the author(s).

---

[1]Here $p_n$ denotes the optimization momentum buffer, which differs by a sign from the physical momentum variable in (2b).

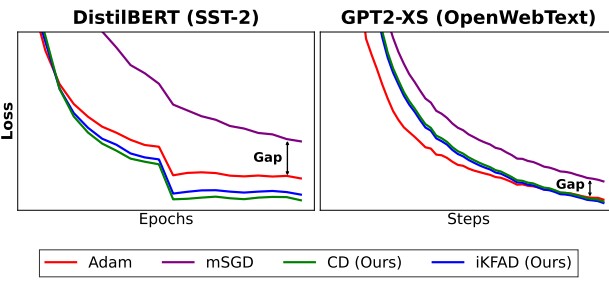

*Figure 1.* **Adam-mSGD gap**: Selected experiments demonstrating that CD and iKFAD can reduce the gap between Adam and mSGD without per-parameter adaptive learning rates on language modeling tasks using Transformers.

Linearly Dissipative Hamiltonian Dynamics (LDHD) (Gao et al., 2022; Maddison et al., 2018; Simsekli et al., 2020):

$$\dot{x} = p, \tag{2a}$$
$$\dot{p} = -\nabla f(x) - \gamma p, \tag{2b}$$

where $\gamma > 0$ is the friction coefficient. The "$-\gamma p$" term introduces linear damping or friction into the dynamics. Conversely, applying an Euler discretization to the system (2) yields the discrete dynamics in (1) suggesting that the linear dissipation manifests as the exponentially weighted averaging of gradients characteristic of classical momentum. This relationship establishes a direct equivalence between the friction coefficient $\gamma$ and the momentum coefficient $\mu$ through the relation

$$\mu = 1 - \gamma\sqrt{\delta t} \tag{3}$$

(see Appendix A). Consequently, the momentum coefficient $\mu$ in the discrete setting performs the exact same role as the friction $\gamma$ in continuous time: both regulate the momenta $p$ and by extension, the kinetic energy of the optimizer.

In continuous time, the optimization process can be viewed as the motion of a particle navigating a potential field defined by the loss function $f(x)$. The optimizer state is defined by its position $x$ and momentum $p$, with the total energy $H(x, p) = f(x) + \frac{1}{2}\|p\|^2$ corresponding to the sum of the current loss (potential energy) and kinetic energy. In the absence of dissipation ($\gamma = 0$), energy is conserved and the dynamics oscillate around the minimum without settling. Dissipation is necessary to remove kinetic energy and allow the dynamics to converge. While the momentum mechanism is often viewed as introducing exponentially weighted averages of past gradients, this Hamiltonian perspective reframes it as a dissipative mechanism that regulates the optimizer dynamics. The above $\mu - \gamma$ equivalence shows that a fixed momentum coefficient $\mu$ corresponds to applying uniform damping across all coordinates, which may be insufficient in highly anisotropic loss landscapes.

Here, it is also useful to present the continuous dynamics of

Adam (Da Silva & Gazeau, 2020), (Karoni, 2024). [2]

$$\dot{x} = \frac{p}{\sqrt{\zeta + \epsilon}}, \tag{4a}$$
$$\dot{p} = -\nabla f(x) - \gamma p, \tag{4b}$$
$$\dot{\zeta} = [\nabla f(x)]^2 - \alpha\zeta, \tag{4c}$$

where "$[\cdot]^k$" denotes element-wise exponentiation. Adam (Kingma & Ba, 2015) combines the momentum mechanism with per-parameter adaptive learning rates. Adam performs this adaptation by maintaining an exponential moving average of squared gradients. While this quantity does not directly estimate curvature, it provides a crude diagonal approximation to the gradient second moment, enabling adaptive per-parameter learning rates that dampen updates in directions with high gradient variability.

Instead of employing per-parameter learning rates to account for anisotropy in the optimization dynamics, our approach relies on coordinate-wise friction coefficients. Our iKFAD and CD methods correspond to replacing the constant friction coefficient in Equation (2b) with a coordinate-wise adaptive friction term. The resulting dynamics are of the following general form

$$\dot{x} = p, \tag{5a}$$
$$\dot{p} = -\nabla f(x) - g(p) \star p, \tag{5b}$$

where "$\star$" denotes element-wise vector multiplication. As we will see, $g(p)$ for iKFAD will be equal to an exponentially weighted average of past kinetic energies, whereas for CD, $g(p)$ will be directly proportional to the instantaneous kinetic energy. As a preliminary demonstration of these dynamics, we examine the behaviour of our optimizers in two low-dimensional examples (see Figures 2 and 3 and observe that our methods are more stable than mGD and exhibit faster convergence in these simple settings.

**Conflict of Interest Disclosure**   The authors declare no financial or other substantive conflicts of interest.

## 2. Methodology

The linear dissipation term $-\gamma p$ in Equation (2b) acts as a damping force on the momenta, with the value of the friction coefficient $\gamma$ directly affecting their magnitude. Insufficient damping (low values of the friction parameter $\gamma$) can lead to high momenta, which can in turn result in oscillations and instability (O'Donoghue & Candès, 2015). Excessive damping on the other hand (high values of $\gamma$), leads to excessive

---

[2]We note that the continuous-time dynamics in Eq. (4) are not intended as an exact continuous limit of the discrete Adam equations (Kingma & Ba, 2015), as they do not include bias correction.

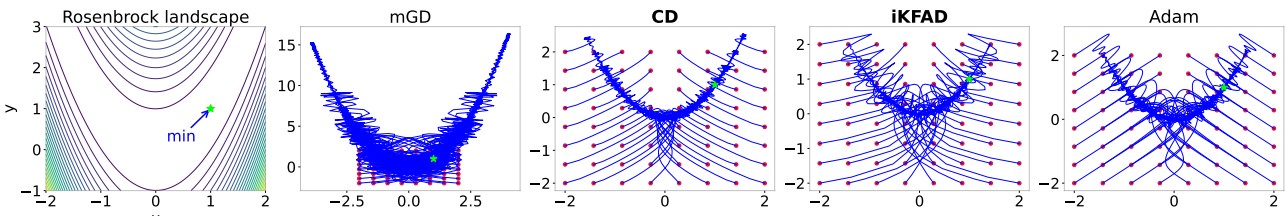

*Figure 2.* **Phase portraits for a two-dimensional Rosenbrock function**: Each red grid point corresponds to a different initialization in the $(x, y)$ domain and each blue line corresponds to a different optimization trajectory. The minimum $(x_{\min}, y_{\min}) = (1, 1)$, is denoted by a green star. We set $\gamma = 1$, $h = 0.005$, and $\alpha = \rho = c = 1$. mGD exhibits significant oscillations and overshooting of the minimum (note the different scale of the mGD y-axis). In contrast, our per-parameter, adaptive friction method iKFAD and cubic damping method CD lead to smoother and much more direct trajectories towards the minimum, similarly to Adam.

energy dissipation, small momenta and slow convergence. Having a friction or momentum coefficient that dynamically adapts based on the kinetic energy of the system would allow us to increase the damping when momenta are getting too high and decrease it when momenta are lower. This acts like a gentle thermostat on the system, allowing us to control the magnitude of the momenta. This adaptive friction mechanism was first introduced in the context of optimization in (Karoni et al., 2023) with only a single adaptive coefficient for all parameters. Here, we modify the original dynamics, by introducing individual adaptive friction coefficients for each of the model parameters.

We refer to this new optimizer as *Individual Kinetic Friction Adaptive Descent (iKFAD)* and describe its dynamics as

$$\dot{x} = p, \tag{6a}$$

$$\dot{p} = -\nabla f(x) - \gamma p - \xi \star p, \tag{6b}$$

$$\dot{\xi} = \frac{[p]^2}{\rho} - \alpha\xi, \tag{6c}$$

where $\gamma, \alpha, \rho > 0$, $x, p, \xi \in \mathbf{R}^N$.

Although Equation (6b) includes a fixed friction $\gamma$, as well as the adaptive friction mechanism, we find that in practice, setting $\gamma = 0$ still maintains good performance. Retaining a fixed friction term, however, is helpful in obtaining theoretical convergence guarantees.

Note that, while in the discrete setting, Adam is viewed as performing per-parameter learning rate adaptation through the second-moment estimate $\zeta$, its continuous time formulation can also be interpreted as performing per-parameter momentum adaptation. This provides a useful parallel with iKFAD. Both methods regulate the momenta $p$ on a per-parameter basis: Adam performs this adaptation in the position equation, $\dot{x} = \dfrac{p}{\sqrt{\zeta + \epsilon}}$, whereas iKFAD introduces this control directly through the momentum equation, $\dot{p} = -\nabla f(x) - \gamma p - \xi \star p$.

To better understand the role of the adaptive friction coefficient $\xi$, we look at the exact solution of equation (6c), given

as

$$\xi(t) = \mathrm{e}{-\alpha t}\xi(0) + \frac{1}{\rho}\int_0^t [p(s)]^2 \mathrm{e}{-\alpha(t-s)}\,\mathrm{d}s.$$

We see that $\xi$ is an exponentially weighted average of the squares of past momenta, i.e. kinetic energies. Thus, a history of high momenta leads to an increased damping coefficient $\xi$, which helps "cool down" the system and control oscillatory behaviour which might otherwise arise in the presence of high momenta. Conversely, a history of low momenta leads to less damping, allowing the optimizer to advance along low gradient, low-momenta regions.

Our individual adaptive friction scheme addresses coordinate-wise anisotropy in the optimization dynamics in a manner analogous to the per-parameter learning rates used in adaptive optimizers such as Adam. While iKFAD does not explicitly rescale the gradient, the use of a separate friction coefficient for each parameter in the momentum update (6b) effectively regulates the update velocity on a per-coordinate basis. This is particularly relevant for architectures like Transformers, where parameters in different layers or attention heads exhibit vastly different update scales and noise characteristics, often limiting the effectiveness of mSGD's global momentum. By introducing parameter-specific damping, iKFAD adaptively modulates the inertia of each parameter, allowing the optimizer to suppress oscillations in high-variance directions while maintaining progress in flatter regions, achieving a similar stabilizing effect to Adam.

Next, we consider the near–equilibrium behaviour of (6c). When $\alpha$ is sufficiently large and upon setting $\dot{\xi} \approx 0$, we obtain

$$\xi \approx (\alpha\rho)^{-1}[p]^2.$$

Substituting this approximation into (6b) yields

$$\dot{p} \approx -\nabla f(x) - (\alpha\rho)^{-1}[p]^3,$$

indicating that iKFAD behaves like a *cubically damped momentum method* in this regime. This motivates the study

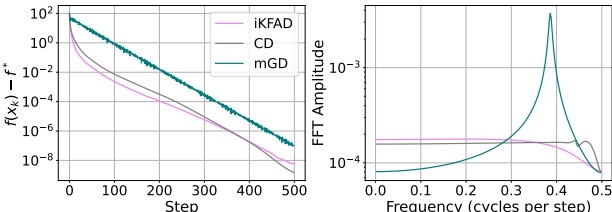

*Figure 3.* **Effect of adaptive friction and cubic damping on a two-hundred dimensional anisotropic quadratic**: $f(\mathbf{x}) = \frac{1}{2}\mathbf{x}^T\mathbf{A}\mathbf{x}$, with eigenvalues ranging between 1 and $10^4$. For momentum gradient descent (mGD), we choose the theoretically optimal learning rate $h = 2/\sqrt{M}$ and friction $\gamma = 2\sqrt{m}$ (Sanz Serna & Zygalakis, 2021), where $m = 1$ and $M = 10^4$ are the minimum and maximum eigenvalues of $A$ respectively. **Left**: Objective function value $f(x)$ across optimization steps. Note that both iKFAD and CD outperform mGD in this setting. **Right**: Fourier spectrum of trajectory in direction of highest eigenvalue.

of the effect of cubic damping in state of the art optimizers such as mSGD and Adam. Cubic damping has long been studied as an effective vibration-suppression technique in engineering applications (Panananda et al., 2012) (Peng et al., 2010)(Zhu & Lang, 2022)(Babister, 1976). Higher odd-order damping (such as quintic damping) would also be possible, provided that we use an odd power, so that the sign of the momenta is maintained.

We first introduce *Cubically Damped mSGD (CD)*, the dynamics of which are obtained by augmenting the momentum equation (2b) in LDHD by a cubic damping term.

$$\dot{x} = p, \tag{7a}$$
$$\dot{p} = -\nabla f(x) - \gamma p - c[p]^3. \tag{7b}$$

An advantage of the cubic damping mechanism is that it is more gentle than its linear counterpart for low momenta and becomes more aggressive than linear damping as momentum increases. This allows for gentler damping and therefore enhanced exploration along directions where gradients and momenta are low, while allowing for increased damping and oscillation control along high-momentum directions.

Figure 3 shows CD and iKFAD converging faster than theoretically optimal mGD on a 200-dimensional anisotropic quadratic function. Interestingly, mGD exhibits a pronounced spectral peak in the direction of the highest eigenvalue, while iKFAD and CD are able to significantly suppress oscillatory modes. Figure 2 shows phase portraits of various optimizers on the Rosenbrock function. For each optimizer, we initialize a grid of starting points (red dots) and plot the corresponding deterministic trajectories (blue lines). CD and iKFAD display strong suppression of oscillations akin to Adam, whereas mGD exhibits significant oscillations and overshooting of the minimum.

We now introduce our third optimizer, *Cubically damped Adam (CADAM)*, the dynamics of which are obtained by augmenting the momentum equation in (4) by a cubic damping term:

$$\dot{x} = \frac{p}{\sqrt{\zeta + \epsilon}}, \tag{8a}$$
$$\dot{p} = -\nabla f(x) - \gamma p - c[p]^3, \tag{8b}$$
$$\dot{\zeta} = [\nabla f(x)]^2 - \alpha\zeta. \tag{8c}$$

To numerically integrate the continuous-time dynamics, we employ an *operator splitting* scheme. The key idea of operator splitting is to decompose the full vector field of the ODE into a sum of simpler sub-operators, each of which can be integrated exactly. The full update over one time step is then obtained by composing these partial flows, a type of splitting method (see (Leimkuhler & Reich, 2005)), although in practice an Euler-type discretization can be used of suitable modification of the dissipation coefficient is used. The explicit update rules for each step are provided in Appendix B.

In terms of memory and compute, CD maintains an identical memory footprint of $2N$ optimizer states (position and momentum) as mSGD, while adding only negligible computational overhead for the element-wise cubing of the momentum vector. Similarly, iKFAD and CADAM have the same memory requirements as Adam ($3N$) as they each maintain one additional auxiliary vector state variable (adaptive friction ($\xi$) for iKFAD and second-moment estimate ($\zeta$) for CADAM).

## 3. Related Work

Although adaptation of the momentum hyperparameter $\mu$ has been suggested in prior works, the area remains relatively unexplored. The authors of (Sutskever et al., 2013) advocated for a gradual increase in the momentum coefficient as training progresses, based on the theoretical convergence results of (Nesterov, 1983; 2013). We provide a comparison of iKFAD against two known momentum schedules on FashionMNIST training with a ResNet18 in Appendix C. The authors of (Su et al., 2016) derive a second-order ODE for Nesterov's accelerated gradient method, which also involves a time-dependent friction coefficient. In (Karoni et al., 2023), an adaptive momentum coefficient based on the kinetic energy of the system was demonstrated (on molecular optimization) to lead to faster convergence and increased robustness with respect to the choice of learning rate compared to the use of a fixed momentum parameter. Adaptive friction has been studied in Langevin sampling, known as the Adaptive Langevin method (Ding et al., 2014; Shang et al., 2015), which employs a thermostat to regulate noisy gradients. Controlling the momenta $p$ to stabilize the

optimization process and avoid oscillations has also been explored in (O'Donoghue & Candès, 2015), where the authors propose a momentum restart mechanism that resets the momenta to zero when oscillatory behaviour emerges, typically once the momenta exceed a critical threshold. Rather than resetting momenta, iKFAD employs a continuous control mechanism that adapts the friction to keep the momenta within a stable range. In terms of theoretical work, (Dobson et al., 2025) prove that choosing the friction parameter based on the smallest eigenvalue of the Hessian leads to convergence speed-ups in the case of strongly convex potentials.

## 4. Theoretical Results

This section establishes the convergence properties of iKFAD and CD. For strongly convex objective functions $f \in C^2$ with bounds on the highest and lowest eigenvalues of the Hessian, we prove that the continuous-time dynamics of iKFAD and CD exhibit exponential convergence to the global minimum. We also demonstrate that this exponential convergence rate is preserved in the discrete-time schemes for a sufficiently small $\delta t$. We leave the convergence analysis of CADAM for future work.

### 4.1. iKFAD Convergence Analysis

**Theorem 4.1.** *Consider a function $f \in C^2$ and assume that there exist $a, b > 0$ such that*

$$
\begin{aligned}
a\big[f(x) - f(x^*)\big] &+ b\|x - x^*\|^2 \\
&\leq (x - x^*) \cdot (\nabla f(x) - \nabla f(x^*)),
\end{aligned} \tag{9}
$$

*where $x^*$ is the unique global minimum of $f$. Then, for any initial condition $(x_0, p_0, \xi_0) \in \mathbf{R}^d \times \mathbf{R}^d \times \mathbf{R}_+^d$, and considering $\gamma > 0$, there exist $\kappa > 0$ and $C \in \mathbf{R}_+$ such that the solution of (6) satisfies*

$$
f(x(t)) - f(x^*) + \|p(t)\| + \|\xi(t)\| \leq C\mathrm{e}^{-\kappa t}.
$$

The condition (9) is satisfied for $f \in C^2$ with $0 < m \leq \nabla^2 f(x) \leq M < +\infty$. A similar condition is considered in (Mattingly et al., 2002).

*Proof.* See Appendix H.2 for the proof, which follows the same reasoning as the FAD continuous convergence proof in (Karoni et al., 2023). □

Consider the splitting (17) of the continuous iKFAD dynamics and the integration scheme CDBA (see Table 3), found in appendices B and H.3 respectively. The resulting discrete

iKFAD dynamics can be written as

$$
p_{n+1} = \beta_{n,\delta t} p_n - \delta t \nabla f(x_n), \tag{10a}
$$

$$
x_{n+1} = x_n + \delta t \beta_{n,\delta t} p_n - \delta t^2 \nabla f(x_n), \tag{10b}
$$

$$
\xi_{n+1} = \mathrm{e}^{-\alpha \delta t}\left([\xi_n]^2 + \frac{(I - \mathrm{e}^{-2\Xi_n \delta t})[p_n]^2}{\rho}\right)^{1/2}, \tag{10c}
$$

where $\Xi = \mathrm{diag}(\xi)$ and $\beta_{n,\delta t} = \mathrm{e}^{-(\gamma I + \Xi_n)\delta t}$ (see Appendix H.3 for more details). We can then state the following convergence result.

**Theorem 4.2.** *Consider a function $f \in C^2$ satisfying $0 < m \leq \nabla^2 f(x) \leq M < +\infty$, for any $x \in \mathbf{R}^d$. Assume that $\gamma > 0$ and consider $L > 0$. Then, for any initial condition $(x_0, p_0, \xi_0) \in \mathbf{R}^d \times \mathbf{R}^d \times \mathbf{R}_+^d$ such that*

$$
\|x_0\| + \|p_0\| + \|\xi_0\| \leq L,
$$

*there exist $\delta t^\star > 0$, $\kappa > 0$ and $C > 0$ (depending on $L$) such that for any $\delta t \in (0, \delta t^\star)$ and any $n \geq 0$,*

$$
f(x_n) - f(x^*) + \|p_n\|^2 + \|\xi_n\|^2 \leq C\mathrm{e}^{-\kappa n \delta t}.
$$

*Proof.* See Appendix H.3. The proof follows the strategy of the FAD discrete convergence analysis in (Karoni et al., 2023). □

### 4.2. CD Convergence Analysis

**Theorem 4.3.** *Consider a function $f \in C^2$. Assume that $\gamma, c > 0$ and that there exist $a, b > 0$ such that*

$$
\begin{aligned}
a\big[f(x) - f(x^*)\big] &+ b\|x - x^*\|^2 \\
&\leq (x - x^*) \cdot (\nabla f(x) - \nabla f(x^*)).
\end{aligned} \tag{11}
$$

*Then, for any initial condition $(x_0, p_0) \in \mathbf{R}^d \times \mathbf{R}^d$, there exist $\kappa > 0$ and $C \in \mathbf{R}_+$ such that the solution of (7) satisfies*

$$
f(x(t)) - f(x^*) + \|p(t)\|^2 \leq C\mathrm{e}^{-\kappa t}.
$$

*Proof.* See Appendix I.2 □

We consider an Euler discretization of (7), which is simpler to work with than the splitting scheme proposed in Table 3:

$$
p_{n+1} = (1 - \gamma \delta t)p_n - c\delta t[p_n]^3 - \delta t \nabla f(x_n), \tag{12a}
$$

$$
x_{n+1} = x_n + \delta t p_n, \tag{12b}
$$

where we assume that $\gamma \delta t$ is sufficiently small.

**Theorem 4.4.** *Consider $f \in C^2$ and assume that there exist $m, M \in \mathbf{R}_+$ such that $m \leq \nabla^2 f(x) \leq M$ for any $x \in \mathbf{R}^d$. Fix $L > 0$. Then, for any initial condition $(x_0, p_0)$ such that $\|x_0\| + \|p_0\| \leq L$, there exist $\delta t^* > 0, r > 0$ and $C > 0$ for which, for any $n \geq 0$ and for any $\delta t \in (0, \delta t^*)$,*

$$
f(x_n) - f(x^*) + \|p_n\|^2 \leq C\mathrm{e}^{-\kappa n \delta t}.
$$

*Proof.* See Appendix I.3 □

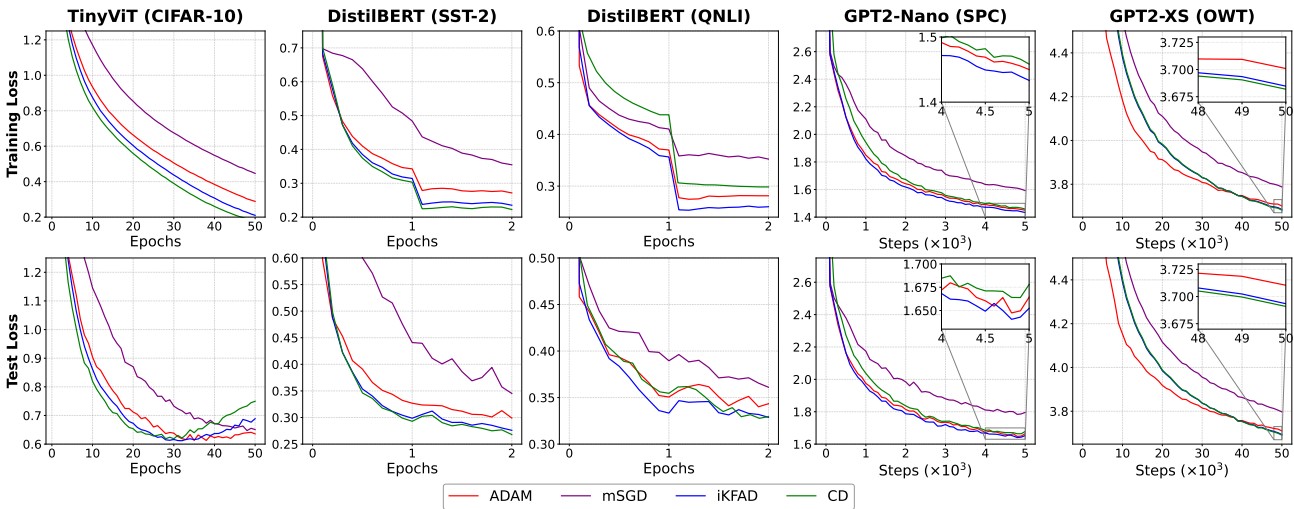

*Figure 4.* **Training and test losses** for iKFAD, CD, Adam, and mSGD across ResNet-18, DistilBERT, and GPT2. All curves are averaged over 10 random seeds. Standard deviations are omitted for readability. Across the transformer-based tasks considered, a clear performance gap is observed between Adam and mSGD. Both iKFAD and CD substantially reduce this gap despite not using per-parameter learning rates.

# 5. Experiments

Our code is available on GitHub.

## 5.1. Main Benchmark Results

To evaluate the performance of our proposed optimizers, we conduct a series of experiments across different machine learning tasks, including image classification on CIFAR-10 (Krizhevsky et al., 2009) using ResNet-18 and TinyViT (Wu et al., 2022), language classification with DistilBERT (66M) for SST-2 GLUE (Wang et al., 2018) and QNLI fine-tuning, and language generation via GPT2-Nano (0.85M) (Karpathy, 2022) as well as a custom GPT2-XS (45M). We downsized GPT2-S for computational efficiency by reducing the depth to 6 layers, the number of heads to 8 and the embedding dimension to 512. We refer to this as GPT2-XS. For language modeling experiments, we utilize a batch size of 16, following evidence that small batches can achieve performance comparable to larger scales (Marek et al., 2025), while TinyViT and ResNet-18 utilize a batch size of 128. We include comparisons against Adam and mSGD (standard implementations). We allocate equal resources to all optimizers during hyperparameter search. For a clean and simpler comparison, we omit common regularization and acceleration techniques such as learning rate scheduling, weight decay, and gradient accumulation. ResNet-18 results are reported in Appendix D, as all optimizers exhibited very similar performance in that setting. CADAM results are also discussed in Appendix E, as we did not observe a significant improvement in performance over Adam across the benchmarks considered. A possible explanation is that cubic damping provides limited additional benefit when

per-parameter learning rates are already being used.

Hyperparameter optimization for all schemes was conducted using Optuna Bayesian search with a fixed budget of 80 trials per experiment, with specific search ranges detailed in Appendix F. Our evaluation includes two distinct sweeps for the proposed optimizers; one with $\gamma > 0$ and one with $\gamma = 0$, selecting the top-performing configuration for each. Notably, CD and iKFAD exhibit similar performance (see Section 5.2). All the descriptions below pertain to Figure 4.

**TinyViT.** iKFAD and CD outperform Adam. They both reduce training loss faster and reach a slightly lower test loss. mSGD converges much more slowly, although it eventually approaches a similar test loss. We also note that the TinyViT accuracies obtained are competitive with those reported in (Wu et al., 2022) (without the use of distillation).

**DistilBERT.** There is a substantial performance gap between Adam and mSGD across both the SST-2 and QNLI tasks. On the SST-2 pretraining task, iKFAD and CD outperform Adam in both training and test loss. For the QNLI finetuning task, while a large gap persists between Adam and mSGD in training loss, the discrepancy in test loss is notably smaller. iKFAD reaches a lower test loss more rapidly, whereas CD initially follows Adam before eventually matching iKFAD's test loss. We observe a sharp decline in training loss at the start of the second epoch, most noticeably in QNLI, which is not mirrored in the validation loss and suggests the onset of overfitting.

**GPT2.** We evaluated GPT2-Nano on Shakespeare and GPT2-XS on OpenWebText, and observe the same pattern

across both benchmarks. The pronounced Adam-mSGD gap is visible in both scenarios.

In GPT2-Nano, iKFAD, CD and Adam exhibit very similar behaviour. For GPT2-XS, the Adam loss decreases slightly faster during early training, but CD and iKFAD eventually reach final test losses similar to that of Adam. The training and test loss trajectories of iKFAD and CD are nearly identical. This similarity is consistent with iKFAD operating near steady state ($\dot{\xi} \approx 0$) for this benchmark.

Across all benchmarks considered, iKFAD, CD, and CADAM consistently outperform mSGD and achieve performance competitive with Adam, while in some experiments attaining slightly lower training and test losses. Table 1 summarizes the best losses averaged over ten runs in each experiment along with the corresponding standard deviations. The accuracies for relevant experiments can be found in Figure 5. The tuned hyperparameters $\rho$ in iKFAD and $c$ in CD exhibit a wide range in magnitude across tasks indicating a strong dependence on gradient scales in each setting. This was not explored, but it could be remedied by normalizing gradients between training iterations.

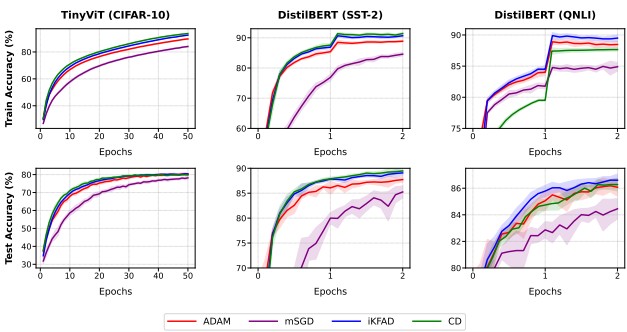

*Figure 5.* Corresponding training and test accuracies for the experiments in Figure 4. Standard deviations are included.

The GPT2-Nano and GPT2-XS models used in the experiments of Figure 4 are downsized models with 0.85M and 45M parameters respectively. To assess whether the observed performance of our methods persists at a larger transformer scale, we additionally train a GPT2-S model (123M) parameters on OpenWebText. As shown in Figure 6, iKFAD and CD remain competitive with Adam and outperform mSGD on this task.

### 5.2. Ablation on Linear Damping

REMOVING LINEAR DAMPING $\gamma = 0$.

To study the role of the linear damping term $\gamma$, we repeat the experiments in Figure 4 with $\gamma = 0$ and present the results in Figure 7. This isolates the effect of adaptive friction and cubic damping. We find that while CADAM tends not to perform very well without linear friction (see Appendix

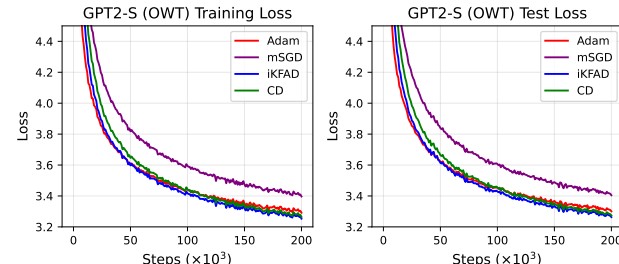

*Figure 6.* **Training and test losses** for iKFAD, CD, Adam, and mSGD on GPT2-S (123M parameters) trained on OpenWebText. The curves correspond to a single random seed due to computational constraints. Our methods iKFAD and CD remain competitive with Adam and achieve lower losses than mSGD.

*Table 1.* Best Test Loss results ($\mu \pm \sigma$).

| Model (Dataset) | Adam | mSGD | CD | iKFAD |
|---|---|---|---|---|
| TinyViT (CIFAR-10) | $0.613 \pm 0.018$ | $0.647 \pm 0.012$ | $0.619 \pm 0.016$ | $\mathbf{0.612 \pm 0.014}$ |
| DistilBERT (SST-2) | $0.299 \pm 0.012$ | $0.345 \pm 0.022$ | $\mathbf{0.268 \pm 0.009}$ | $0.276 \pm 0.011$ |
| DistilBERT (QNLI) | $0.340 \pm 0.012$ | $0.361 \pm 0.016$ | $\mathbf{0.328 \pm 0.010}$ | $0.329 \pm 0.013$ |
| GPT2-Nano (SPC) | $1.647 \pm 0.010$ | $1.784 \pm 0.011$ | $1.664 \pm 0.008$ | $\mathbf{1.641 \pm 0.006}$ |
| GPT2-XS (OWT) | $3.710 \pm 0.011$ | $3.797 \pm 0.008$ | $\mathbf{3.691 \pm 0.016}$ | $3.693 \pm 0.018$ |
| GPT2-S (OWT) | $3.3008$ | $3.4047$ | $3.2736$ | $\mathbf{3.2633}$ |

E Figure 12), CD and iKFAD maintain their good performance even when $\gamma = 0$. This is expected because after hyperparameter optimization in the $\gamma > 0$ setting, the optimal $\gamma$ values were typically near the lower end of the search range for CD and iKFAD. Hence, we can set $\gamma = 0$, which reduces the hyperparameter count for both CD and iKFAD to two and three hyperparameters respectively (matching mSGD and Adam). We note that the hyperparameter search space for sweeps with $\gamma = 0$ searches over one dimension less than if we set $\gamma > 0$. This offers a possible explanation as to why $\gamma = 0$ sweeps arrive at a better hyperparameter configuration. The results of CADAM are presented in Appendix E. We observe that CADAM tends to rely more heavily on linear damping (see Figure 12 and Table 5 where optimal $\gamma$ values corresponding to CADAM are significantly higher than for iKFAD and CD).

VARYING $\gamma$ AND $\delta t$.

To assess the robustness of iKFAD and CD with respect to the learning rate $\delta t$ and linear friction $\gamma$, we evaluated both optimizers across the NanoGPT (SPC), DistilBERT (SST2), and TinyViT (CIFAR-10) tasks. Note that while we vary $\gamma$ and $\delta t$, the rest of the hyperparameters are kept fixed at some reasonable values (the best setting for each experiment from Section 5). We compare against mSGD and LDHD. For mSGD, we map $\gamma$ to the momentum parameter $\mu$ using the correspondence derived in Appendix A. LDHD corresponds to the same continuous dynamics as mSGD, but while mSGD uses an Euler discretization, LDHD uses a splitting discretization. Figure 8 shows that iKFAD and CD maintain substantially higher robustness and accuracy over

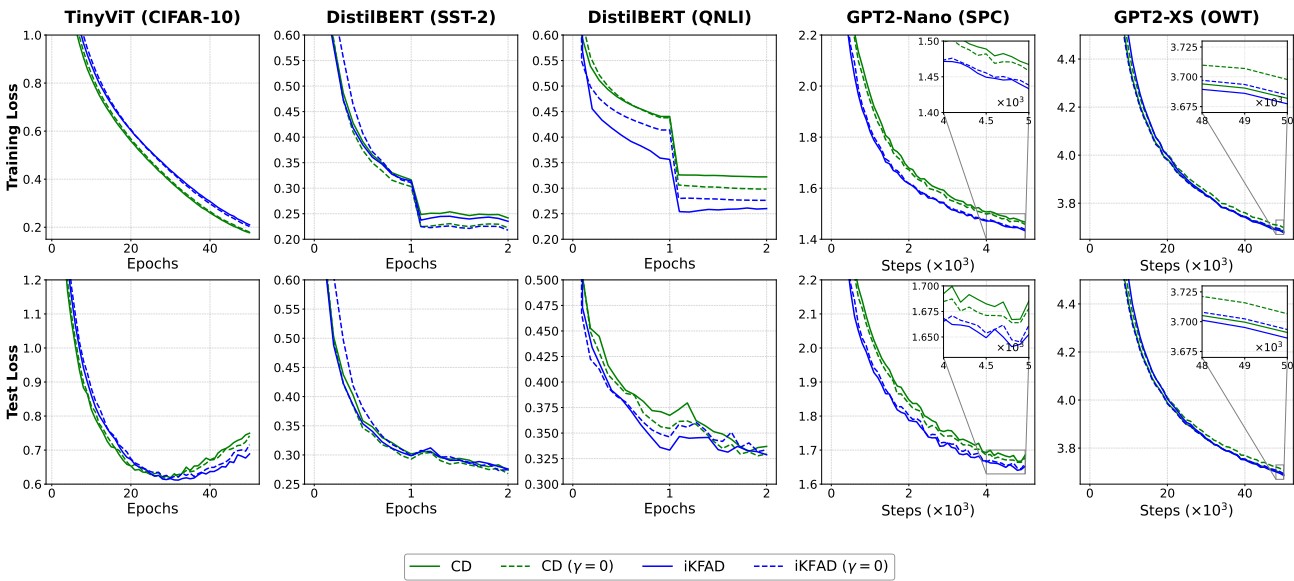

*Figure 7.* **Removing linear damping** ($\gamma = 0$). Sweeping CD and iKFAD hyperparameters with $\gamma = 0$ versus $\gamma \in [10^{-6}, 10]$ shows little sensitivity to $\gamma$. This suggests $\gamma$ can be set to zero in practice, reducing the number of hyperparameters by one. The best CD and iKFAD results from the current figure (either $\gamma = 0$ or $\gamma > 0$) were selected to present in Figure 4.

wide ranges of $(\gamma, \delta t)$ compared to mSGD and LDHD. Both iKFAD and CD also exhibit very similar behaviour in terms of test metrics. This indicates that iKFAD is close to equilibrium $\dot{\xi} = 0$ for the majority of training. This is expected, as iKFAD and CD are closely related: in both methods the amount of damping depends on kinetic energy. In iKFAD, the adaptive friction variable $\xi$ is an exponentially weighted average of past kinetic energies ($[p]^2$), so damping depends on a short history of the dynamics. In CD, the damping is instantaneous. The cubic term can be written as $-c[p]^2 \star p$, i.e., the damping coefficient scales with the current kinetic energy. We note that both optimizers remain effective even for very small $\gamma$, demonstrating that their adaptive friction and cubic damping mechanisms can compensate for minimal linear damping. We also observe that choices of $c$ close to the heuristic correspondence $c \approx 1/(\alpha\rho)$ perform well in some experiments, such as NanoGPT, which reflects the aforementioned connection between the two methods. As a side note, mSGD and LDHD share the same underlying continuous dynamics, but their stability regions differ under the discretizations used, as demonstrated in Figure 8.

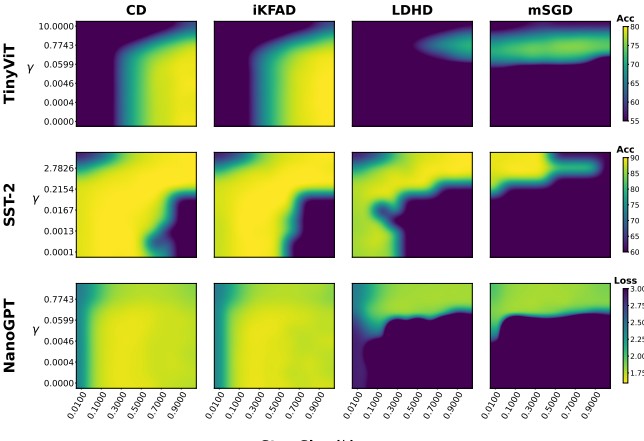

*Figure 8.* $\gamma - \delta t$ grid. Test accuracies/losses for iKFAD, CD, LDHD, and mSGD on various experiments as a function of learning rate $\delta t$ and linear friction $\gamma$. Higher color intensity indicates better performance. $\gamma$ for mSGD is obtained by the correspondence derived in Appendix A. LDHD uses splitting for discretization.

### 5.3. iKFAD versus KFAD: Importance of Individual Frictions

To demonstrate the significance of individual friction parameters, we perform a comparison between KFAD and iKFAD on TinyViT and NanoGPT. KFAD corresponds to the original FAD formulation (Karoni et al., 2023) which employs a single adaptive global friction coefficient. We observe that although KFAD outperforms mSGD, it performs worse than iKFAD and CD (see Figure 9 and Table 2), indicating that parameter-wise friction adaptation is important

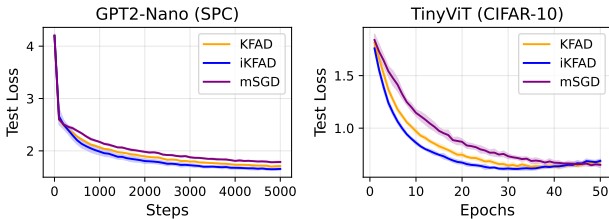

*Figure 9.* Training and test losses for KFAD, iKFAD and mSGD.

*Table 2.* Best Test Loss results ($\mu \pm \sigma$).

| Model (Dataset) | iKFAD | KFAD |
|---|---|---|
| TinyViT (CIFAR-10) | **0.612 ± 0.014** | $0.6424 \pm 0.014$ |
| GPT2-Nano (SPC) | **1.641 ± 0.006** | $1.7363 \pm 0.018$ |

for large machine learning tasks, especially for transformer architectures.

Table 2 summarizes the best test losses (mean and standard deviation, averaged over ten runs). KFAD hyperparameters were optimized using the same Bayesian search sweep budget and same hyperparameter ranges as iKFAD (see Table 5). Note that while for Tiny-ViT, the best KFAD and iKFAD losses are close, iKFAD reaches this loss value substantially faster (see Figure 9).

These results show that while the global adaptive friction method KFAD improves over mSGD, introducing individual adaptive friction parameters leads to further significant improvement, with iKFAD achieving performance much closer to Adam as seen in previous experiments.

## 6. Conclusion

In this work, we introduce Individual Kinetic Friction Adaptive Descent (iKFAD), Cubically Damped mSGD (CD), and Cubically Damped Adam (CADAM), three optimizers motivated by a continuous-time view of momentum-based optimization as a dissipative dynamical system, in which friction regulates kinetic energy. The connection between adaptive friction and cubic damping shows that both iKFAD and CD can be understood as nonlinear damping mechanisms with kinetic energy-dependent friction coefficients. Our theoretical analysis establishes exponential convergence for both continuous-time dynamics of iKFAD and CD as well as the corresponding discrete-time schemes.

Across several transformer-based vision and language benchmarks, including TinyViT, DistilBERT, and GPT2 experiments, our methods achieve performance competitive with and in some cases, better than Adam, while substantially reducing the Adam–mSGD gap on the tasks considered. In particular, iKFAD and CD achieve this without relying on per-parameter adaptive learning rates. Notably,

CD (with $\gamma = 0$) maintains the same memory footprint and number of hyperparameters as standard mSGD while achieving performance competitive with Adam on several transformer benchmarks.

**Limitations and Future Work.** The computational cost of hyperparameter tuning limited our study to at most 80 trials per experiment. Also, since our benchmarks were limited to models under 100M parameters, evaluating these schemes on large-scale LLMs remains future work. On the theory side, we have not yet obtained exponential convergence guarantees for CADAM similar to those proved for iKFAD and CD.

## Acknowledgements

Part of this research was supported by the MAC-MIGS Centre for Doctoral Training (AK) and the ProbAI Hub (AK and RR). The work of AK was also supported by IBM Research. The work of GS was supported by Hi! PARIS and ANR/France 2030 program (ANR-23-IACL-0005). We would like to thank Cameron Barker for his useful insights on configurations for language model training. The authors acknowledge the use of resources provided by the Isambard-AI National AI Research Resource (AIRR) (McIntosh-Smith et al., 2024). Isambard-AI is operated by the University of Bristol and is funded by the UK Government's Department for Science, Innovation and Technology (DSIT) via UK Research and Innovation; and the Science and Technology Facilities Council [ST/AIRR/I-A-I/u6ih].

## Impact Statement

This paper presents work whose goal is to advance the field of Machine Learning. There are many potential societal consequences of our work, none of which we feel must be specifically highlighted here.

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

## A. Equivalence between mSGD and LDHD

In this section we will show how the mSGD discrete equations (1a), (1b) can be obtained as an Euler discretisation of linearly dissipative Hamiltonian dynamics (LDHD) (2a), (2b). We start with the system of ODEs corresponding to LDHD

$$\dot{x} = p,$$
$$\dot{p} = -\nabla f(x) - \gamma p.$$

Applying Euler discretization, with time step $\delta t$, we obtain

$$
\begin{aligned}
p_{n+1} &= p_n + \delta t(-\nabla f(x_n) - \gamma p_n), \\
x_{n+1} &= x_n + \delta t p_{n+1},
\end{aligned}
\quad\Longrightarrow\quad
\begin{aligned}
p_{n+1} &= (1 - \delta t\gamma)p_n - \delta t \nabla f(x_n), \\
x_{n+1} &= x_n + \delta t p_{n+1}.
\end{aligned}
$$

Dividing the momentum equation by the learning rate $\delta t$, we obtain

$$
\begin{aligned}
\frac{p_{n+1}}{\delta t} &= (1 - \delta t\gamma)\frac{p_n}{\delta t} - \nabla f(x_n), \\
x_{n+1} &= x_n + \delta t^2 \frac{p_{n+1}}{\delta t},
\end{aligned}
\quad\overset{\tilde{p} = p/\delta t}{\Longrightarrow}\quad
\begin{aligned}
\tilde{p}_{n+1} &= (1 - \delta t\gamma)\tilde{p}_n - \nabla f(x_n), \\
x_{n+1} &= x_n + \delta t^2 \tilde{p}_{n+1}.
\end{aligned}
$$

Finally, after setting $\tilde{\delta t} = \delta t^2$ and $\mu = 1 - \gamma\sqrt{\tilde{\delta t}}$, we have

$$
\begin{aligned}
\tilde{p}_{n+1} &= \left(1 - \gamma\sqrt{\tilde{\delta t}}\right)\tilde{p}_n - \nabla f(x_n), \\
x_{n+1} &= x_n + \tilde{\delta t}\tilde{p}_{n+1},
\end{aligned}
\quad\overset{\bar{p} = -\tilde{p}}{\Longrightarrow}\quad
\begin{aligned}
\bar{p}_{n+1} &= \mu\bar{p}_n + \nabla f(x_n), \\
x_{n+1} &= x_n - \tilde{\delta t}\bar{p}_{n+1},
\end{aligned}
$$

which is exactly the system of discrete momentum gradient descent equations (1a), (1b). Although this equivalence is correct, there are pairings $(\gamma, \delta t)$ for which it would be invalid because $\mu > 0$ *i.e.*, $\gamma\delta t > 1$.

## B. Update Rules for Splitting

To derive practical discrete-time algorithms from the continuous-time dynamics, we use an operator-splitting approach. The idea is to decompose a complex system of ordinary differential equations (ODEs) into a sum of simpler sub-dynamics, for which explicit sub-step updates can be derived. Rather than integrating the full coupled system at once, which typically admits no analytical solution, we sequentially apply these updates over short time intervals and compose them to obtain a single update step.

*Table 3.* Splitting of ODE dynamics for iKFAD, CD, and CADAM.

| | |
|---|---|
| **iKFAD** | $\begin{pmatrix}\dot{x}\\\dot{p}\\\dot{\xi}\end{pmatrix} = \underbrace{\begin{pmatrix}p\\0\\0\end{pmatrix}}_{A} + \underbrace{\begin{pmatrix}0\\-\nabla f(x)\\0\end{pmatrix}}_{B} + \underbrace{\begin{pmatrix}0\\-\xi\star p\\\frac{[p]^2}{\rho}\end{pmatrix}}_{C} + \underbrace{\begin{pmatrix}0\\-\gamma p\\-\alpha\xi\end{pmatrix}}_{D}$ |
| **CD** | $\begin{pmatrix}\dot{x}\\\dot{p}\end{pmatrix} = \underbrace{\begin{pmatrix}p\\0\end{pmatrix}}_{A} + \underbrace{\begin{pmatrix}0\\-\nabla f(x)\end{pmatrix}}_{B} + \underbrace{\begin{pmatrix}0\\-c[p]^3\end{pmatrix}}_{C'} + \underbrace{\begin{pmatrix}0\\-\gamma p\end{pmatrix}}_{D'}$ |
| **CADAM** | $\begin{pmatrix}\dot{x}\\\dot{p}\\\dot{\zeta}\end{pmatrix} = \underbrace{\begin{pmatrix}\frac{p}{\sqrt{\zeta+\epsilon}}\\0\\0\end{pmatrix}}_{A'} + \underbrace{\begin{pmatrix}0\\-\nabla f(x)\\0\end{pmatrix}}_{B} + \underbrace{\begin{pmatrix}0\\-c[p]^3\\0\end{pmatrix}}_{C'} + \underbrace{\begin{pmatrix}0\\-\gamma p\\0\end{pmatrix}}_{D'} + \underbrace{\begin{pmatrix}0\\0\\[\nabla f(x)]^2 - \alpha\zeta\end{pmatrix}}_{E}$ |

Concretely, consider an ODE of the form $\dot{z} = \sum_k F_k(z)$, where each vector field $F_k$ represents a distinct part of the dynamics, such as transport, gradient forcing, damping, or adaptive scaling. Operator splitting approximates the flow of

the full system over a time step $\delta t$ by composing sub-step updates associated with each $F_k$. When these sub-steps have closed-form solutions, this gives explicit update rules without requiring numerical ODE solvers. For background on splitting methods, see (Leimkuhler & Reich, 2005).

Table 3 lists the resulting decomposition into sub-operators (labeled A–E), while Table 4 reports the update rules used for each sub-step over a time interval $\delta t$. While steps A, A', B, C', D, D' and E admit analytical updates, the update for step C is obtained by observing that the quantity $p_i^2 + \rho\xi_i^2$ is invariant (Karoni et al., 2023). The discrete-time optimizers studied in the main text are obtained by composing these analytical updates in a prescribed order.

The discrete-time optimizers are constructed using first-order Lie-Trotter operator splitting. Specifically, iKFAD uses the CDBA composition, CD follows a C'D'BA sequence and CADAM uses a C'D'BAE composition. Higher-order symmetric splittings, such as the second-order Strang splitting, could also be used, but these first-order compositions give simple explicit schemes with low computational cost per iteration. In each case, the full update $z_{n+1} = \Phi_{\delta t}(z_n)$ is obtained by the sequential application of the sub-step updates detailed in Table 4.

Table 4. Analytical updates for unique splitting steps.

| Step | Analytical update |
|---|---|
| A | $x_{n+1} = x_n + \delta t\, p_n$ |
| A' | $x_{n+1} = x_n + \delta t\, \frac{p_n}{\sqrt{\zeta_n + \epsilon}}$ |
| B | $p_{n+1} = p_n - \delta t\, \nabla f(x_n)$ |
| C | $p_{n+1} = \mathrm{e}^{-\Xi_n \delta t} p_n,$ $\tilde{\xi}_{n+1} = \sqrt{[\xi_n]^2 + \frac{(I - e^{-2\Xi_n \delta t})}{\rho}[p_n]^2}$ |
| C' | $p_{n+1} = \frac{p_n}{\sqrt{1 + 2c[p_n]^2 \delta t}}$ |
| D | $p_{n+1} = \mathrm{e}^{-\gamma \delta t} p_n,$ $\xi_{n+1} = \mathrm{e}^{-\alpha \delta t}\tilde{\xi}_{n+1}$ |
| D' | $p_{n+1} = \mathrm{e}^{-\gamma \delta t} p_n$ |
| E | $\zeta_{n+1} = \mathrm{e} - \alpha \delta t \zeta_n + \frac{(1 - e^{-\alpha \delta t})}{\alpha}[\nabla f(x_n)]^2$ |

The C' update in Table 4 follows by solving the scalar equation $\dot{p}_i = -cp_i^3$ component-wise. Separating variables and integrating yields

$$\int_{p_i(t)}^{p_i(t+\delta t)} \frac{dq_i}{q_i^3} = -c \int_t^{t+\delta t} ds \Rightarrow \left[-\frac{1}{2q_i^2}\right]_{p_i(t)}^{p_i(t+\delta t)} = -c\,\delta t \Rightarrow p_i(t+\delta t) = \frac{p_i(t)}{\sqrt{1 + 2cp_i(t)^2\delta t}}.$$

## C. Comparison of Known Momentum Schedules to our Adaptive Scedule

Both (Sutskever et al., 2013) and (Nesterov, 1983) propose gradually increasing the momentum parameter $\mu$ for non-strongly-convex functions, which is typical of neural network loss landscapes. The schedules are given by

$$\mu_t = \min\left(1 - 2^{-1-\log_2(\lfloor \frac{t}{250}\rfloor + 1)},\ \mu_{\max}\right),$$
$$\mu_t = 1 - \frac{3}{t+5}, \tag{13}$$

respectively. iKFAD naturally reproduces this behaviour. Since an increase in $\mu$ corresponds to a decrease in $\gamma$ ((3)), the gradual reduction of friction observed in Figure 10 aligns with the progressive momentum increase proposed in (Nesterov, 1983) and (Sutskever et al., 2013). As optimization progresses toward the equilibrium $(\nabla f(x^*), p^*) = (0, 0)$, the system's kinetic energy decays, leading to a corresponding decrease in the adaptive friction $\xi$. Intuitively, higher friction (lower momentum) at the beginning of training allows careful exploration of the loss landscape, while lower friction later facilitates traversal of flat regions and local minima. Unlike static or monotonic schedules (Sutskever et al., 2013; Nesterov, 1983),

our thermostated mechanism adapts friction dynamically and individually per parameter, preventing excessive buildup of momentum.

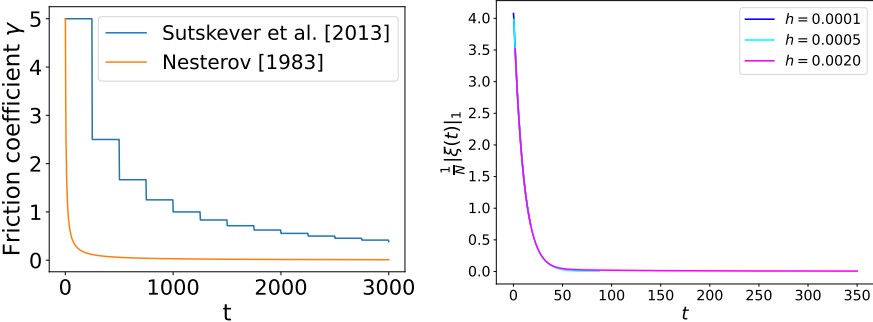

*Figure 10.* Left: Friction schedules corresponding to (13) via the mapping in (3) ($h = 0.01$). Right: Mean $L^1$-norm absolute component value averaged over the number of network parameters) of the adaptive friction $\xi$ in iKFAD (with $\gamma = 0$) during ResNet18 training on FashionMNIST for three learning rates. The decreasing trend of $\xi$ mirrors the momentum annealing behaviour in classical schedules.

## D. ResNet-18 on CIFAR-10 Results

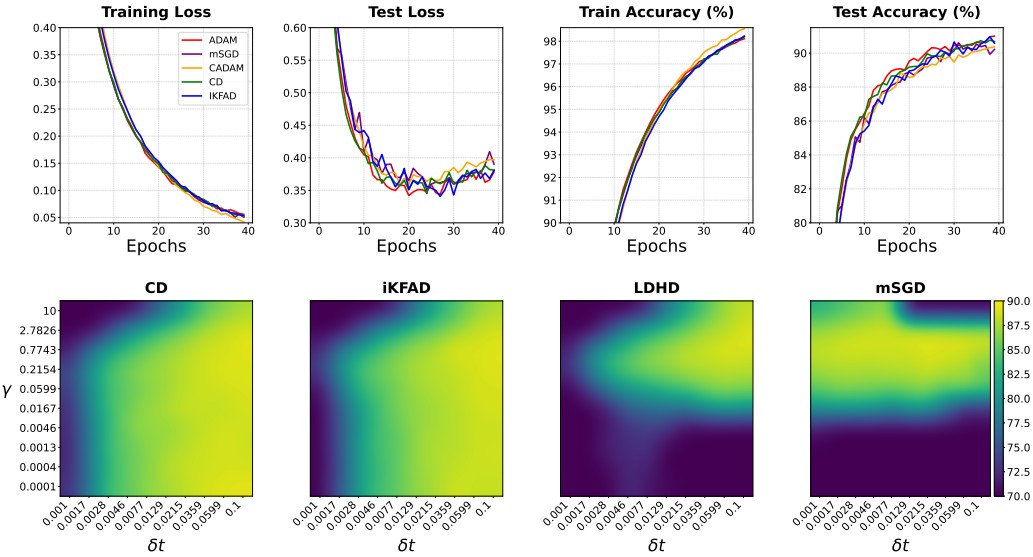

*Figure 11.* Complete Results on ResNet-18. First row shows the train and test accuracy and losses. The second row reports accuracy for different combinations of learning rate and linear damping coefficient.

For ResNet-18, we train on CIFAR-10 with a batch size of 128. After hyperparameter tuning, the performance of all optimizers is comparable. This is consistent with previous empirical observations that, after sufficient hyperparameter tuning, standard optimizers often achieve similar performance on ResNet and VGG architectures (Choi et al., 2020). However, iKFAD and CD appear to be more robust with respect to the choice of linear friction $\gamma$ and step size $\delta t$, as can be seen in the second row of Figure 11. The comparison between LDHD and mSGD highlights the effect of discretization. Both methods correspond to the same underlying continuous dynamics and only differ in the way they are discretized. It appears that they are mostly stable in the same regions, but mSGD performs worse in the high $\gamma$, high $\delta t$ regime.

# E. CADAM Results

This section extends the results of Figure 4 by including the corresponding CADAM results and examines the effect of removing the linear damping term in CADAM. Like Adam, CADAM maintains three state variables, but introduces an extra hyperparameter, the cubic damping coefficient $c$. In Figure 12, we study the effect of removing the linear damping mechanism and find that doing so leads to a substantial decline in performance across all benchmarks considered. These results suggest that the linear damping term plays an important stabilizing role in CADAM. When $\gamma = 0$, the method relies entirely on cubic damping to control the momentum variables, which appears to be insufficient for these benchmarks. In particular, the GPT2-Nano training loss diverges for $\gamma = 0$.

When linear damping is included, CADAM exhibits substantially improved performance. Figure 13 extends Figure 4 by adding the corresponding CADAM results and standard deviations across random seeds. We observe that CADAM performance is comparable to Adam on several of the studied benchmarks, as shown in Figure 13. However, CADAM does not provide a consistent advantage over Adam across the tasks considered. In particular, on TinyViT, CADAM converges more slowly than Adam (although final losses are very close) whereas for the DistilBERT model trained on SST-2, CADAM achieves a lower average training and test loss than Adam. For the remaining tasks, CADAM performance is broadly comparable that of Adam.

Overall, our results suggest that augmenting the Adam dynamics with a cubic damping term provides limited additional benefit in our setting. One potential explanation for this could be that adaptive friction and cubic damping play a role similar to Adam's adaptive learning rates. As discussed in Section 2, both Adam and iKFAD can be interpreted in continuous time as regulating the momentum variables on a per-parameter basis, although through different equations. Moreover, the cubic damping term can be viewed as involving a nonlinear, momentum-dependent friction coefficient $g(p)$ within the general framework of Equation (5). Since Adam already includes coordinate-wise adaptive scaling through $\zeta$, adding cubic damping may introduce a partially redundant form of adaptivity, which could explain why CADAM does not consistently improve over Adam in our experiments.

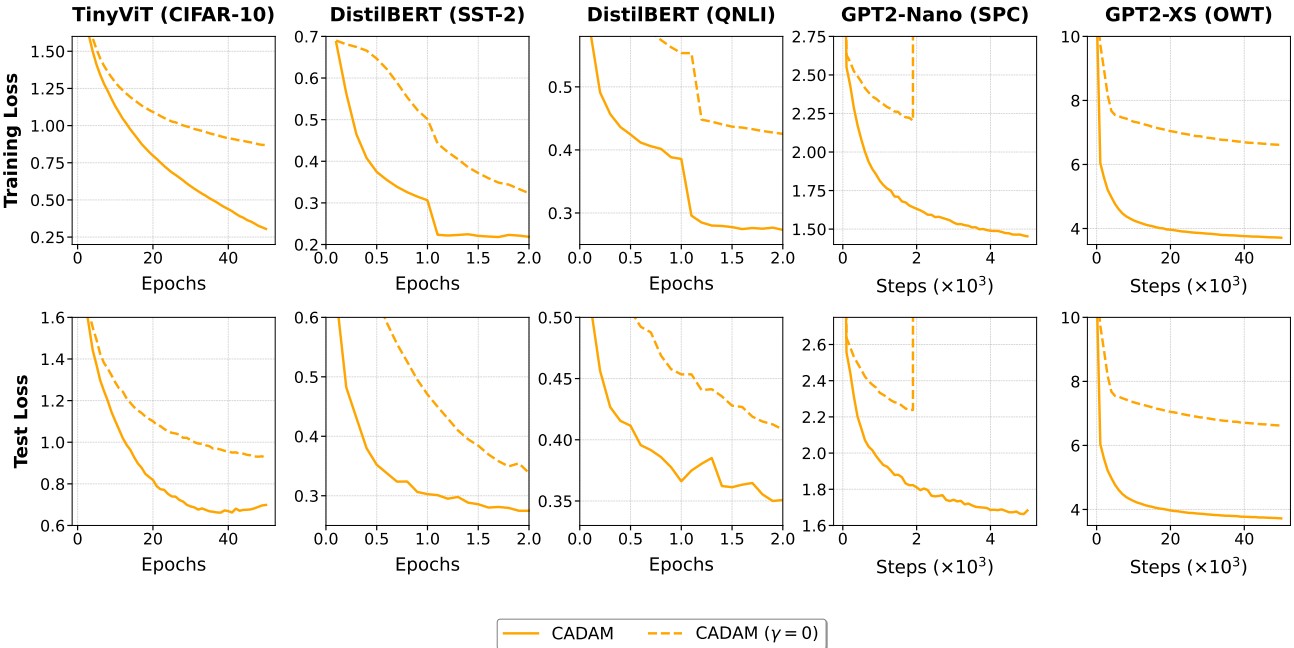

*Figure 12.* **CADAM with and without linear damping.** Training and test losses for CADAM with $\gamma = 0$ and with $\gamma \in [10^{-6}, 10]$ across the benchmark tasks. Removing the linear damping term leads to worse performance across the benchmarks considered, indicating that linear damping plays an important role in CADAM.

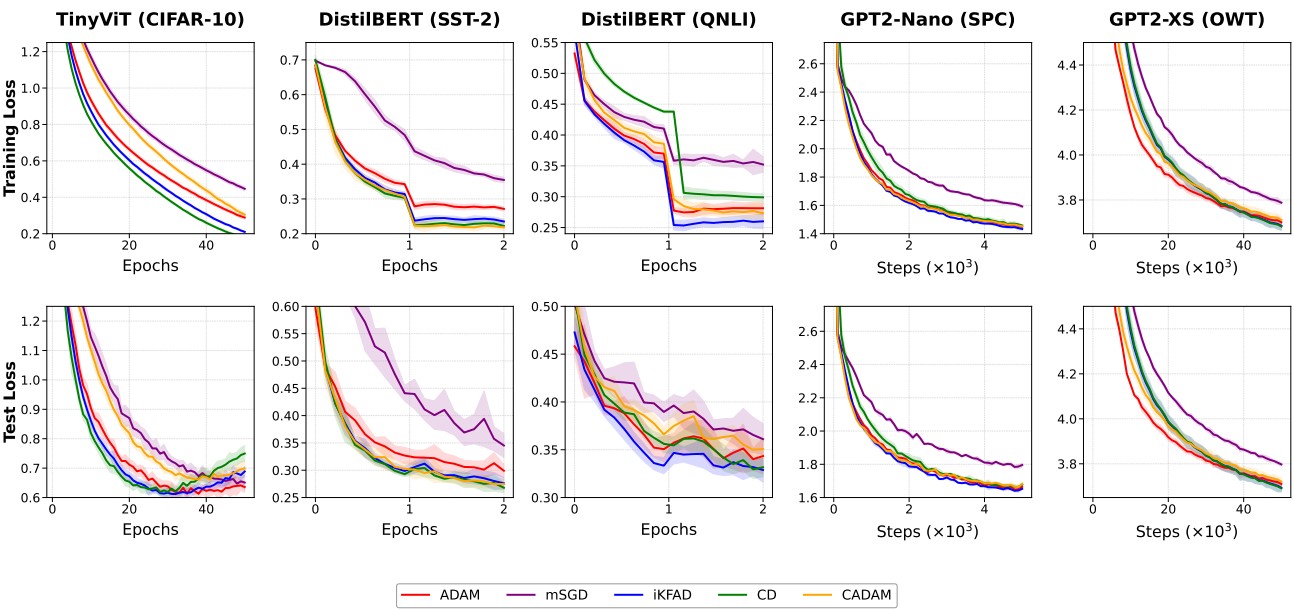

*Figure 13.* **Training and test losses including CADAM.** Training and test loss curves for CADAM, Adam, mSGD, iKFAD and CD across benchmarks. Curves are averaged over random seeds, with shaded regions indicating standard deviations. CADAM achieves performance broadly comparable to Adam on several tasks, but does not provide a consistent improvement over Adam in these experiments.

# F. Hyperparameter Values for Optuna Searches

Across models and tasks, the best-performing hyperparameter scales for $\rho$ (iKFAD) and $c$ (CD) varied by several orders of magnitude. This variation is likely related to the scale of the momentum variables during training, since these coefficients control the strength of the nonlinear damping terms. Normalizing the momentum variables across iterations could potentially reduce this sensitivity to the magnitude of $p$, but we leave this direction for future work.

Experimental configurations:

- **ResNet18-CIFAR-10**: batch size 128, 40 epochs, $\gamma \in [10^{-9}, 10]$, $(\rho, c, \alpha) \in [10^{-9}, 10^{10}]$. $h \in [10^{-6}, 10^{-2}]$ for CADAM and $h \in [10^{-6}, 10^{-1}]$ for iKFAD and CD. The ranges here are quite vast as this was the very first experiment.

- **TinyViT - CIFAR-10**: batch size 128, 50 epochs. $\gamma \in [10^{-5}, 10]$, $\rho \in [10^{-8}, 10]$, $c \in [10^4, 10^{10}]$. $h \in [10^{-6}, 10^{-2}]$ for CADAM and $h \in [10^{-4}, 3 \times 10^{-1}]$ for iKFAD and CD.

- **DistilBERT - SST-2**: batch size 16, 2 epochs, $\gamma \in [10^{-6}, 1]$ for CADAM/iKFAD and $\gamma = 0$ for CD, $\rho \in [10^{-8}, 10]$, $\alpha \in [0.001, 1]$.

- **DistilBERT - QNLI**: batch size 16, 2 epochs, $\gamma \in [10^{-6}, 1]$ for CADAM/iKFAD and $\gamma = 0$ CD, $c \in [0.01, 10^{13}]$, $\alpha \in [0.001, 10]$, $\rho \in [10^{-8}, 10]$.

- **GPT2-Nano - Shakespeare**: batch size 16, 5000 steps. $\gamma \in [10^{-5}, 10]$, $h \in [10^{-5}, 5 \times 10^{-1}]$ for CD/iKFAD and $h \in [10^{-6}, 10^{-2}]$ for CADAM, $c \in [10^{-1}, 10^6]$, $\alpha \in [10^{-3}, 10]$ for CADAM and $\alpha \in [10^{-5}, 10]$ for iKFAD. $\rho \in [10^{-8}, 10]$

- **GPT2-XS - OpenWebText**: batch size 16, 50000 steps. $h \in [10^{-6}, 10^{-2}]$ for CADAM and $h \in [10^{-4}, 10^{-1}]$ for CD/iKFAD. $\gamma \in [10^{-5}, 10]$. $\alpha \in [10^{-3}, 1]$. $c \in [10^4, 10^{10}]$. $\rho \in [10^{-8}, 10]$.

- **GPT2-S - OpenWebText**: batch size 16, 200000 steps. Same hyperparameter ranges as GPT2-XS.

For all experiments, hyperparameters for Adam and mSGD were swept over the following ranges: For Adam, we used $h \in [10^{-6}, 10^{-2}]$, $\beta_1 \in [0.85, 0.999]$ and $\beta_2 \in [0.85, 0.999]$. For mSGD, we used $h \in [10^{-6}, 10^{-1}]$ for mSGD and $\mu \in [0.5, 0.99]$.

*Table 5.* Optimized hyperparameters for all loss curves in Figure 4.

| | $\delta t$ | $\gamma$ | $\alpha$ | $\rho$ | $c$ | $\beta_1$ | $\beta_2$ | $\mu$ |
|---|---|---|---|---|---|---|---|---|
| **ResNet-18** | | | | | | | | |
| iKFAD | 0.0973 | 0.40 | 0.402 | 1.87 | - | - | - | - |
| CD | 0.0448 | 0.17 | - | - | 56814.47 | - | - | - |
| CADAM | 0.0037 | 3.55 | 0.005 | - | 82552.90 | - | - | - |
| mSGD | 0.0260 | - | - | - | - | - | - | 0.86 |
| Adam | 0.0008 | - | - | - | - | 0.97 | 0.99 | - |
| **TinyViT** | | | | | | | | |
| iKFAD | 0.08991 | 0.04537 | 0.10110 | $6.29 \times 10^{-6}$ | - | - | - | - |
| CD | 0.19983 | 0.00221 | - | - | $6.55 \times 10^5$ | - | - | - |
| CADAM | 0.00197 | 4.46870 | 24314309 | - | 0.91377 | - | - | - |
| mSGD | 0.02133 | - | - | - | - | - | - | 0.87439 |
| Adam | 0.00055 | - | - | - | - | 0.86082 | 0.88521 | - |
| **SST-2** (Pretrain) | | | | | | | | |
| iKFAD | 0.02704 | $1.28 \times 10^{-6}$ | 0.0465 | $3.61 \times 10^{-7}$ | - | - | - | - |
| CD | 0.0363 | 0 | - | - | $9.78 \times 10^6$ | - | - | - |
| CADAM | 0.00035 | 9.2188 | 0.03233 | - | 0.0237 | - | - | - |
| mSGD | 0.00237 | - | - | - | - | - | - | 0.8335 |
| Adam | $3.82 \times 10^{-5}$ | - | - | - | - | 0.9187 | 0.98254 | - |
| **QNLI** (Finetune) | | | | | | | | |
| iKFAD | 0.0413 | $1.36 \times 10^{-8}$ | 2.80 | 6.71 | - | - | - | - |
| CD | 0.03144 | 0 | - | - | $1.90 \times 10^8$ | - | - | - |
| CADAM | 0.0002 | 9.66 | 0.0026 | - | 22754.57 | - | - | - |
| mSGD | 0.0015 | - | - | - | - | - | - | 0.895 |
| Adam | $3.52 \times 10^{-5}$ | - | - | - | - | 0.855 | 0.933 | - |
| **GPT2-XS** (OWT) | | | | | | | | |
| iKFAD | 0.09956 | 0. | 0.04756 | $1.04 \times 10^{-5}$ | - | - | - | - |
| CD | 0.09975 | 0.00138 | - | - | $1.43 \times 10^6$ | - | - | - |
| CADAM | 0.00678 | 7.53239 | 0.44020 | - | $3.11 \times 10^6$ | - | - | - |
| mSGD | 0.05362 | - | - | - | - | - | - | 0.95277 |
| Adam | 0.00065 | - | - | - | - | 0.89447 | 0.99454 | - |
| **GPT2-S** (OWT) | | | | | | | | |
| iKFAD | 0.0937 | 0. | 0.1933 | $4.2726 \times 10^{-7}$ | - | - | - | - |
| CD | 0.0629 | 0. | - | - | $11.8506 \times 10^6$ | - | - | - |
| CADAM | - | - | - | - | - | - | - | - |
| mSGD | 0.0768 | - | - | - | - | - | - | 0.8746 |
| Adam | $1.0478 \times 10^{-4}$ | - | - | - | - | 0.9001 | 0.9971 | - |
| **NanoGPT** (Shakespeare) | | | | | | | | |
| iKFAD | 0.39055 | $2.58 \times 10^{-5}$ | 0.03474 | 0.00017 | - | - | - | - |
| CD | 0.42614 | 0 | - | - | $1.95 \times 10^5$ | - | - | - |
| CADAM | 0.00828 | 9.47376 | 8.82877 | - | 0.95201 | - | - | - |
| mSGD | 0.09791 | - | - | - | - | - | - | 0.90054 |
| Adam | 0.00168 | - | - | - | - | 0.88757 | 0.92653 | - |

## G. Ablation of $\gamma = 0$

For the $\gamma = 0$ ablation studies, we used the same hyperparameter search ranges as in the previous section, but fixed $\gamma$ to zero. In some cases, the best runs with $\gamma = 0$ performed better than the corresponding best best runs with $\gamma > 0$. These runs were therefore used in Figure 4 as the overall best. For CD and iKFAD, the best values of $\delta t$, $\alpha$, and $\rho/c$ are generally of comparable scale across the $\gamma > 0$ and $\gamma = 0$ search settings, as shown in Table 6.

*Table 6.* Comparison of optimal hyperparameters. Left: Unconstrained ($\gamma > 0$). Right: Constrained ($\gamma = 0$). Note: The $\rho/c$ column denotes $\rho$ for iKFAD and $c$ for CD/CADAM.

| Dataset | Method | Unconstrained ($\gamma > 0$) | | | | Constrained ($\gamma = 0$) | | |
|---|---|---|---|---|---|---|---|---|
| | | $\delta t$ | $\gamma$ | $\alpha$ | $\rho/c$ | $\delta t$ | $\alpha$ | $\rho/c$ |
| **TinyViT** | CD | 0.200 | 0.0022 | – | $6.55 \times 10^5$ | 0.117 | – | $5.35 \times 10^5$ |
| | iKFAD | 0.090 | 0.045 | 0.101 | $6.29 \times 10^{-6}$ | 0.073 | 0.090 | $1.26 \times 10^{-5}$ |
| | CADAM | 0.0020 | 4.47 | 0.914 | $2.43 \times 10^7$ | $8.17 \times 10^{-4}$ | 0.173 | $8.84 \times 10^9$ |
| **SST-2** | CD | 0.0422 | 0.052 | – | $3.34 \times 10^7$ | 0.0363 | – | $9.78 \times 10^6$ |
| | iKFAD | 0.0270 | $1.28 \times 10^{-6}$ | 0.047 | $3.61 \times 10^{-7}$ | 0.0165 | 0.057 | $9.89 \times 10^{-6}$ |
| | CADAM | $3.49 \times 10^{-4}$ | 9.22 | 0.032 | 0.024 | $3.39 \times 10^{-5}$ | 0.021 | $3.39 \times 10^5$ |
| **QNLI** | CD | 0.0789 | 2.78 | – | $1.12 \times 10^8$ | 0.0314 | – | $1.90 \times 10^8$ |
| | iKFAD | 0.0413 | $1.36 \times 10^{-8}$ | 2.80 | $6.71 \times 10^{-9}$ | 0.0448 | 0.0017 | $2.77 \times 10^{-7}$ |
| | CADAM | $1.60 \times 10^{-4}$ | 9.66 | 0.0026 | $2.28 \times 10^4$ | $1.74 \times 10^{-5}$ | 0.033 | $1.52 \times 10^7$ |
| **GPT2-XS** | CD | 0.0997 | 0.0014 | – | $1.43 \times 10^6$ | 0.0990 | – | $1.37 \times 10^6$ |
| | iKFAD | 0.0763 | 0.0009 | 0.6737 | $1.3857 \times 10^{-6}$ | 0.0996 | 0.0476 | $1.0429 \times 10^{-5}$ |
| | CADAM | 0.0068 | 7.53 | 0.440 | $3.11 \times 10^6$ | $1.32 \times 10^{-5}$ | 0.491 | $1.43 \times 10^8$ |
| **NanoGPT** | CD | 0.499 | 0.0038 | – | $1.87 \times 10^5$ | 0.426 | – | $1.95 \times 10^5$ |
| | iKFAD | 0.391 | $2.58 \times 10^{-5}$ | 0.035 | $1.71 \times 10^{-4}$ | 0.494 | 2.20 | $4.55 \times 10^{-6}$ |
| | CADAM | 0.0083 | 9.47 | 8.83 | 0.952 | 0.0052 | 0.012 | $7.02 \times 10^5$ |

## H. iKFAD Convergence Analysis

This section establishes the convergence properties of the proposed iKFAD optimizer. We focus our analysis on strictly convex objective functions (specifically $C^2$ functions with Hessian eigenvalues bounded above and below by positive constants) to provide stability guarantees. We first prove that the continuous-time iKFAD dynamics converges at an exponential rate to the global minimum. We then show that this geometric convergence rate is preserved in a discrete-time setting, provided the step size $\delta t$ is sufficiently small.

### H.1. Dynamical Properties of iKFAD

Equation (6c) reads $\dot{\xi} = \frac{[p(s)]^2}{\rho} - \alpha\xi$. Therefore,

$$\xi(t) = e^{-\alpha t}\xi(0) + \int_0^t e^{-\alpha(t-s)} \frac{[p(s)]^2}{\rho} ds.$$

This means that $\xi(t) \geq 0$ for all $t > 0$, given $\xi(0) \geq 0$. Therefore $\xi$ can be seen as a friction coefficient, similarly to $\gamma$. We next show that under some conditions on $f$ the dynamics of (6) is well-posed on infinite time horizons.

**Lemma H.1.** *Assume that $f$ is smooth and $f(x) \to +\infty$ as $\|x\| \to +\infty$. Then, for any initial condition $(x_0, p_0, \xi_0) \in \mathbf{R}^d \times \mathbf{R}^d \times \mathbf{R}_+^d$, the solution of (6) is well defined for all times $t \geq 0$, and there exists $R > 0$ such that*

$$\forall t \geq 0, \qquad \|x(t)\| \leq R, \quad \|p(t)\| \leq R, \qquad \|\xi(t)\| \leq R.$$

*Proof.* The Cauchy–Lipschitz theorem ensures the existence and uniqueness of a solution for a positive time. To show that

the solution is global in time, we introduce the following Lyapunov function

$$\mathcal{G}(x, p, \xi) = f(x) - f(x^*) + \frac{1}{2}\|p\|^2 + \frac{\rho}{2}\|\xi\|^2.$$

The function $\mathscr{G}(t) = \mathcal{G}(x(t), p(t), \xi(t))$ satisfies

$$\begin{aligned}
\dot{\mathscr{G}}(t) &= \nabla f(x(t)) \cdot \dot{x}(t) + p(t) \cdot \dot{p}(t) + \rho\xi(t) \cdot \dot{\xi}(t) \\
&= -\gamma\|p(t)\|^2 - (\xi(t) \star p(t)) \cdot p(t) + \xi(t) \cdot [p(t)]^2 - \alpha\rho\|\xi(t)\|^2 \\
&= -\gamma\|p(t)\|^2 - \alpha\rho\|\xi(t)\|^2 \le 0,
\end{aligned}$$

where we used that $\xi(t) \ge 0$. This shows that $\mathscr{G}(t) \le \mathscr{G}(0)$, from which the result easily follows since $f(x) - f(x^*) \ge 0$, for all $x \in \mathbf{R}^d$. $\qquad\square$

The next result, whose derivation is straightforward, characterizes equilibria of the dynamics.

**Proposition H.2.** *The equilibria of the system* (6) *correspond to*

$$(\dot{x}, \dot{p}, \dot{\xi}) = 0 \Leftrightarrow (p^*, -\nabla f(x^*), \xi^*) = 0,$$

*i.e. they coincide with the physical equilibria.*

### H.2. Exponential Convergence of the Continuous iKFAD Dynamics

Next, we show exponential convergence of the iKFAD dynamics (6) to the equilibria of Proposition H.2.

**Theorem 4.1.** *Consider a function $f \in C^2$ and assume that there exist $a, b > 0$ such that*

$$\begin{aligned}
a\big[f(x) - f(x^*)\big] &+ b\|x - x^*\|^2 \\
&\le (x - x^*) \cdot (\nabla f(x) - \nabla f(x^*)),
\end{aligned} \tag{9}$$

*where $x^*$ is the unique global minimum of $f$. Then, for any initial condition $(x_0, p_0, \xi_0) \in \mathbf{R}^d \times \mathbf{R}^d \times \mathbf{R}_+^d$, and considering $\gamma > 0$, there exist $\kappa > 0$ and $C \in \mathbf{R}_+$ such that the solution of* (6) *satisfies*

$$f(x(t)) - f(x^*) + \|p(t)\| + \|\xi(t)\| \le Ce^{-\kappa t}.$$

*Proof.* Consider $\varepsilon > 0$ and introduce the following Lyapunov function (which is, up to the additional term $\|\xi\|^2$, a common choice for stochastic Langevin dynamics (Mattingly et al., 2002), also considered for dissipated Hamiltonian dynamics (Moucer et al., 2023)):

$$\mathcal{W}_\varepsilon(x, p, \xi) = f(x) - f(x^*) + \frac{1}{2}\|p\|^2 + \frac{\rho}{2}\|\xi\|^2 + \varepsilon(x - x^*) \cdot p + \varepsilon\|x - x^*\|^2. \tag{14}$$

We first derive a lower bound for $\mathcal{W}_\varepsilon(x, p, \xi)$ which ensures positivity of the Lyapunov function:

$$\mathcal{W}_\varepsilon(x, p, \xi) = f(x) - f(x^*) + \frac{\rho}{2}\|\xi\|^2 + \frac{1}{4}\|p\|^2 + \left\|\frac{p}{2} + \varepsilon(x - x^*)\right\|^2 + \varepsilon(1 - \varepsilon)\|x - x^*\|^2.$$

Assuming $\varepsilon \in (0, 1/2]$, we then have the following lower bound for the Lyapunov function (14):

$$\mathcal{W}_\varepsilon(x, p, \xi) \ge f(x) - f(x^*) + \frac{1}{4}\|p\|^2 + \frac{\rho}{2}\|\xi\|^2 + \frac{\varepsilon}{2}\|x - x^*\|^2. \tag{15}$$

We can similarly derive an upper bound for $\mathcal{W}_\varepsilon(x, p, \xi)$ as follows:

$$\mathcal{W}_\varepsilon(x, p, \xi) = f(x) - f(x^*) + \frac{\rho}{2}\|\xi\|^2 + \frac{3}{4}\|p\|^2 - \left\|\frac{p}{4} - \varepsilon(x - x^*)\right\|^2 + \varepsilon(1 + \varepsilon)\|x - x^*\|^2.$$

As we have assumed $\varepsilon \in (0, 1/2]$, we then have the following upper bound:

$$\mathcal{W}_\varepsilon(x, p, \xi) \leq f(x) - f(x^*) + \frac{3}{4}\|p\|^2 + \frac{\rho}{2}\|\xi\|^2 + \frac{3\varepsilon}{2}\|x - x^*\|^2. \tag{16}$$

We can then upper bound the time derivative of $\mathscr{W}_\varepsilon(t) = \mathcal{W}_\varepsilon(x(t), p(t), \xi(t))$ as follows:

$$
\begin{aligned}
\dot{\mathscr{W}}_\varepsilon(t) &= \nabla f(x(t)) \cdot \dot{x}(t) + p(t) \cdot \dot{p}(t) + \rho \xi(t) \cdot \dot{\xi}(t) + \varepsilon p(t) \cdot \dot{x}(t) \\
&\quad + \varepsilon \dot{p}(t) \cdot (x(t) - x^*) + 2\varepsilon \dot{x}(t) \cdot (x(t) - x^*) \\
&= -(\gamma - \varepsilon)\|p(t)\|^2 - \alpha\rho\|\xi(t)\|^2 - \varepsilon(x(t) - x^*) \cdot \nabla f(x(t)) \\
&\quad - \varepsilon(\gamma - 2)(x(t) - x^*) \cdot p(t) - \varepsilon(x(t) - x^*) \cdot (\xi(t) \star p(t)) \\
&\leq -(\gamma - \varepsilon)\|p(t)\|^2 - \alpha\rho\|\xi(t)\|^2 - \varepsilon(x(t) - x^*) \cdot \nabla f(x(t)) \\
&\quad + \varepsilon |\gamma - 2| \left( \eta\|x(t) - x^*\|^2 + \frac{1}{4\eta}\|p(t)\|^2 \right) + \varepsilon \left( \Delta\|x(t) - x^*\|^2 + \frac{R^2}{4\Delta}\|p(t)\|^2 \right),
\end{aligned}
$$

where we used weighted Young's inequalities as well as Lemma H.1. Specifically, we used that $\|\xi(t)\| \leq R$ so that $\xi_i(t)^2 \leq R^2$ in particular, and therefore $\|\xi(t) \star p(t)\|^2 = \sum_{i=1}^d (\xi_i(t)p_i(t))^2 \leq R^2\|p(t)\|^2$. Using assumption (9), we have

$$
\begin{aligned}
\dot{\mathscr{W}}_\varepsilon(t) &\leq -\left[ \gamma - \varepsilon\left( 1 - \frac{R^2}{4\Delta} - \frac{|\gamma - 2|}{4\eta} \right) \right] \|p(t)\|^2 - \alpha\rho\|\xi(t)\|^2 - a\varepsilon(f(x) - f(x^*)) \\
&\quad - \varepsilon\left( b - |\gamma - 2|\eta - \Delta \right)\|x(t) - x^*\|^2.
\end{aligned}
$$

We set $\eta = \frac{b}{4|\gamma - 2|}$ and $\Delta = b/4$, upon which, we obtain

$$
\begin{aligned}
\dot{\mathscr{W}}_\varepsilon(t) &\leq -\left[ \gamma - \varepsilon\left( 1 - \frac{R^2 + (\gamma - 2)^2}{b} \right) \right] \|p(t)\|^2 - \alpha\rho\|\xi(t)\|^2 - a\varepsilon(f(x) - f(x^*)) \\
&\quad - \frac{\varepsilon b}{2}\|x(t) - x^*\|^2.
\end{aligned}
$$

We choose a sufficiently small $\varepsilon > 0$, so that $\gamma - \varepsilon\left( 1 - \frac{R^2 + (\gamma - 2)^2}{b} \right) > 0$. Using (16), we obtain the following upper bound for the time derivative of the Lyapunov function (14)

$$\dot{\mathscr{W}}_\varepsilon(t) \leq -\min\left\{ \frac{4}{3}\left[ \gamma - \varepsilon\left( 1 - \frac{R^2 + (\gamma - 2)^2}{b} \right) \right], 2\alpha, a\varepsilon, \frac{b}{3} \right\} \mathscr{W}_\varepsilon(t).$$

The above upper bound gives us the desired exponential convergence result through the use of Grönwall's inequality. $\quad\square$

### H.3. Discrete Convergence Analysis

Consider the following splitting of the continuous dynamics (6)

$$
\begin{pmatrix} \dot{x} \\ \dot{p} \\ \dot{\xi} \end{pmatrix} = \underbrace{\begin{pmatrix} p \\ 0 \\ 0 \end{pmatrix}}_{A} + \underbrace{\begin{pmatrix} 0 \\ -\nabla f(x) \\ 0 \end{pmatrix}}_{B} + \underbrace{\begin{pmatrix} 0 \\ -\xi \star p \\ \frac{[p]^2}{\rho} \end{pmatrix}}_{C'} + \underbrace{\begin{pmatrix} 0 \\ -\gamma p \\ -\alpha\xi \end{pmatrix}}_{D}, \tag{17}
$$

and the integration scheme $C'DBA$. For step $C'$, similarly to the analysis performed in (Karoni et al., 2023), we observe that $p_i^2 + \rho\xi_i^2$ is an invariant quantity. Using this, and defining $\Xi = \mathrm{diag}(\xi)$, the integration scheme $C'DBA$ can be written

as follows:

$$\text{Step} \quad C' \qquad \tilde{p}_{n+1/2} = e^{-\Xi_n \delta t} p_n \tag{18}$$

$$\tilde{\xi}_{n+1} = \sqrt{[\xi_n]^2 + \frac{(I - e^{-2\Xi_n \Delta t})}{\rho}[p_n]^2} \tag{19}$$

$$\text{Step} \quad D \qquad \tilde{p}_{n+1} = e^{-\gamma \delta t} \tilde{p}_{n+1/2} \tag{20}$$

$$\xi_{n+1} = e^{-\alpha \delta t} \tilde{\xi}_{n+1} \tag{21}$$

$$\text{Step} \quad B \qquad p_{n+1} = \tilde{p}_{n+1} - \delta t \nabla f(x_n) \tag{22}$$

$$\text{Step} \quad A \qquad x_{n+1} = x_n + \delta t p_{n+1} \tag{23}$$

where for the $C'$ step, we first fix $\xi$ and analytically solve $\dot{p}_i = -\xi_{n,i} p_i \Rightarrow p_{n+1,i} = p_{n,i} e^{-\xi_{n,i} t}$, which, in vector form, corresponds to Equation (18). We then update $\xi$ by observing that for the $C'$ step $\frac{d}{dt}\left(p_i^2 + \rho \xi_i^2\right) = 2 p_i \dot{p}_i + 2\rho \xi_i \dot{\xi}_i = 2 p_i(-\xi_i p_i) + 2\rho \xi_i \frac{p_i^2}{\rho} = 0$. We therefore have $p_{n+1,i}^2 + \rho \xi_{n+1,i}^2 = p_{n,i}^2 + \rho \xi_{n,i}^2$ so that $\xi_{n+1,i}^2 = \xi_{n,i}^2 + \frac{p_{n,i}^2 - p_{n+1,i}^2}{\rho}$. This, written in vector form, yields the update rule for $\xi$ in Equation (19).

Upon defining $\beta_{n,\delta t} = e^{-(\gamma I + \Xi_n)\delta t}$, we can rewrite equations (18)-(23) as

$$p_{n+1} = \beta_{n,\delta t} p_n - \delta t \nabla f(x_n), \tag{24a}$$

$$x_{n+1} = x_n + \delta t \beta_{n,\delta t} p_n - \delta t^2 \nabla f(x_n), \tag{24b}$$

$$\xi_{n+1} = e^{-\alpha \delta t} \sqrt{[\xi_n]^2 + \frac{(I - e^{-2\Xi_n \delta t})[p_n]^2}{\rho}}. \tag{24c}$$

Equations (24) correspond to the system of equations (10). We now show that every equilibrium point of the discrete dynamics is also an equilibrium point of the continuous dynamics, thus the discretization does not introduce any "artificial" equilibrium points. Conversely, we also show that every equilibrium of the continuous dynamics is also an equilibrium of the discrete dynamics.

**Proposition H.3.** *A state $s = (x, p, \xi)$ is an equilibrium point of the discrete iKFAD dynamics given by equations (10a)-(10c) if and only if it is an equilibrium of the continuous iKFAD dynamics (6) .*

*Proof.* Let $s^*$ be an equilibrium of the continuous dynamics as described in Proposition H.2 and let $s_n = (x_n, p_n, \xi_n)$ be an equilibrium point of the discrete dynamics (10a)-(10c). This implies that if we take a step of the discrete dynamics starting from $s_n$, we will remain at equilibrium, that is $s_{n+1} = s_n$. Using equation (23) and setting $x_{n+1} = x_n$ gives us $p_{n+1} = 0$ which implies $p_n = p_{n+1} = 0$. Substituting $p_n = 0$ into Equation (24c) and using $\xi_{n+1} = \xi_n$ yields $\xi_n = e^{-\alpha \delta t} \xi_n$, so that $\xi_n = 0$ as $\alpha \delta t > 0$. Finally, for $p_{n+1} = p_n = 0$, equation (10a) yields $\nabla f(x_n) = 0$.

We next show that any equilibrium point of the continuous dynamics is also an equilibrium of the discrete dynamics. Consider an equilibrium $s^* = (x^*, p^*, \xi^*) = (x^*, 0, 0)$ such that $\nabla f(x^*) = 0$. Let $s_n = s^*$ and let us take a step following the discrete dynamics as given by equations (10a)-(10c). It is straightforward to see that $z_n = (x^*, 0, 0)$ leads to $s_{n+1} = s_n = s^*$. □

Next, we establish convergence to the minimizer for the discrete-time dynamics in Equations (24).

**Theorem 4.2.** *Consider a function $f \in C^2$ satisfying $0 < m \le \nabla^2 f(x) \le M < +\infty$, for any $x \in \mathbf{R}^d$. Assume that $\gamma > 0$ and consider $L > 0$. Then, for any initial condition $(x_0, p_0, \xi_0) \in \mathbf{R}^d \times \mathbf{R}^d \times \mathbf{R}_+^d$ such that*

$$\|x_0\| + \|p_0\| + \|\xi_0\| \le L,$$

*there exist $\delta t^* > 0$, $\kappa > 0$ and $C > 0$ (depending on L) such that for any $\delta t \in (0, \delta t^*)$ and any $n \ge 0$,*

$$f(x_n) - f(x^*) + \|p_n\|^2 + \|\xi_n\|^2 \le C e^{-\kappa n \delta t}.$$

*Proof.* This proof follows the same steps as the convergence analysis of the discretized FAD dynamics presented in (Karoni et al., 2023), the only difference being the fact that $\xi$ in the iKFAD dynamics is a vector, rather than a scalar (as was the case for FAD), and that the matrix $A(x)$ involved in the FAD dynamics is now taken to be the identity $A(x) = I$. Without loss of generality (upon translating and adding a constant to $f(x)$) we can assume that $f(x^*) = 0$ and $x^* = 0$. We consider the Lyapunov function

$$\mathcal{W}(x, p, \xi) = f(x) + \frac{1}{2}\|p\|^2 + A_{\delta t}\rho\|\xi\|^2 + B_{\delta t}\|x\|^2 + C_{\delta t}x \cdot p, \tag{25}$$

which is the Lyapunov function of Equation (14), with coefficients $A_{\delta t}, B_{\delta t}, C_{\delta t}$ to be determined. We can rewrite Equation (25) as

$$\mathcal{W}(x, p, \xi) = f(x) + A_{\delta t}\rho\|\xi\|^2 + \begin{bmatrix} x & p \end{bmatrix} \begin{bmatrix} B_{\delta t} & C_{\delta t}/2 \\ C_{\delta t}/2 & 1/2 \end{bmatrix} \begin{bmatrix} x \\ p \end{bmatrix}.$$

To ensure positivity of $\mathcal{W}(x, p, \xi)$, we need $B_{\delta t} > 0$ and $\det\left(\begin{bmatrix} B_{\delta t} & C_{\delta t}/2 \\ C_{\delta t}/2 & 1/2 \end{bmatrix}\right) > 0$, which translates into $B_{\delta t} > C_{\delta t}^2/2$. We also want $A_{\delta t} > 0$. We start by expressing each of the terms in the Lyapunov function at step $n + 1$ as a function of corresponding terms at step $n$ and deriving upper bounds where possible. Using the fact that $\nabla^2 f(x) \leq M$ we obtain the following upper bound for $f(x_{n+1})$:

$$
\begin{aligned}
f(x_{n+1}) &\leq f(x_n) + \nabla f(x_n) \cdot (x_{n+1} - x_n) + \frac{M}{2}\|x_{n+1} - x_n\|^2 \\
&\leq f(x_n) + \delta t\beta_{n,\delta t}p_n \cdot \nabla f(x_n) + C\delta t^2(\|p_n\|^2 + \|\nabla f(x_n)\|^2).
\end{aligned}
$$

Next,

$$
\begin{aligned}
\|x_{n+1}\|^2 &= \|x_n + \delta t\beta_{n,\delta t}p_n - \delta t^2\nabla f(x_n)\|^2 \\
&\leq \|x_n\|^2 + 2\delta t\beta_{n,\delta t}x_n \cdot p_n - 2\delta t^2 x_n \cdot \nabla f(x_n) + C\delta t^2(\|p_n\|^2 + \|\nabla f(x_n)\|^2).
\end{aligned}
$$

We also have

$$
\begin{aligned}
\|p_{n+1}\|^2 &= \|\beta_{n,\delta t}p_n - \delta t\nabla f(x_n)\|^2 \\
&= p_n \cdot \beta_{n,2\delta t}p_n - 2\delta t\beta_{n,\delta t}p_n \cdot \nabla f(x_n) + \delta t^2\|\nabla f(x_n)\|^2.
\end{aligned}
$$

Moreover,

$$
\begin{aligned}
x_{n+1} \cdot p_{n+1} &= \beta_{n,\delta t}p_n \cdot x_n - \delta t x_n \cdot \nabla f(x_n) + \delta t\|\beta_{n,\delta t}p_n\|^2 \\
&\quad - 2\delta t^2\beta_{n,\delta t}p_n \cdot \nabla f(x_n) + \delta t^3\|\nabla f(x_n)\|^2 \\
&\leq x_n \cdot p_n - \delta t x_n \cdot \nabla f(x_n) - (I - \beta_{n,\delta t})x_n \cdot p_n \\
&\quad + \delta t p_n \cdot \beta_{n,2\delta t}p_n + C\delta t^2(\|p_n\|^2 + \|\nabla f(x_n)\|^2).
\end{aligned}
$$

Finally, we have

$$\rho\|\xi_{n+1}\|^2 = e^{-2\alpha\delta t}[p_n^T(I - e^{-2\Xi_n\delta t})p_n + \rho\|\xi_n\|^2].$$

Given the above, we have the following upper bound for the Lyapunov function at step $n + 1$:

$$
\begin{aligned}
\mathcal{W}(x_{n+1}, p_{n+1}, \xi_{n+1}) &\leq \mathcal{W}(x_n, p_n, \xi_n) - A_{\delta t}\left(1 - e^{-2\alpha\delta t}\right)\rho\|\xi_n\|^2 \\
&\quad + \left[(2B_{\delta t}\delta t\beta_{n,\delta t} - C_{\delta t}(I - \beta_{n,\delta t}))p_n\right] \cdot x_n - (C_{\delta t} + 2B_{\delta t}\delta t)\delta t\, x_n \cdot \nabla f(x_n) \\
&\quad - \left[\frac{1}{2} - A_{\delta t}e^{-2\alpha\delta t}\right]\|p_n\|^2 + \left[\frac{e^{-2\gamma\delta t}}{2} - A_{\delta t}e^{-2\alpha\delta t} + C_{\delta t}\delta t e^{-2\gamma\delta t}\right]p_n^T e^{-2\Xi_n\delta t}p_n \\
&\quad + C\delta t^2\left(\|p_n\|^2 + \|\nabla f(x_n)\|^2\right).
\end{aligned}
$$

We choose $A_{\delta t} = \frac{1}{2}e^{2(\alpha-\gamma)\delta t}$, so that $\frac{e^{-2\gamma\delta t}}{2} - A_{\delta t}e^{-2\alpha\delta t} = 0$. Using $p_n^T e^{-2\Xi_n\delta t}p_n \leq \|p_n\|^2$, we obtain

$$\mathcal{W}(x_{n+1}, p_{n+1}, \xi_{n+1}) \leq \mathcal{W}(x_n, p_n, \xi_n) - \frac{\mathrm{e}^{2(\alpha-\gamma)\delta t}}{2} \left(1 - \mathrm{e}^{-2\alpha\delta t}\right) \rho \|\xi_n\|^2$$
$$+ \left[(2B_{\delta t}\delta t \beta_{n,\delta t} - C_{\delta t}(I - \beta_{n,\delta t})) \, p_n\right] \cdot x_n - (C_{\delta t} + 2B_{\delta t}\delta t)\delta t \, x_n \cdot \nabla f(x_n)$$
$$- \left[\frac{1 - \mathrm{e}^{-2\gamma\delta t}}{2} - C_{\delta t}\delta t \mathrm{e}^{-2\gamma\delta t}\right] \|p_n\|^2 + C\delta t^2 \left(\|p_n\|^2 + \|\nabla f(x_n)\|^2\right).$$

Setting $B_{\delta t} = C_{\delta t} = \varepsilon > 0$, choosing $\varepsilon, \delta t$ to be sufficiently small and using that $x \cdot \nabla f(x) \geq 0$, we obtain

$$\mathcal{W}(x_{n+1}, p_{n+1}, \xi_{n+1}) \leq \mathcal{W}(x_n, p_n, \xi_n) - \frac{\alpha\rho\delta t}{2} \|\xi_n\|^2 - \frac{\gamma\delta t}{2} \|p_n\|^2 - \varepsilon\delta t \, x_n \cdot \nabla f(x_n)$$
$$+ \varepsilon \left([2\delta t \beta_{n,\delta t} - (I - \beta_{n,\delta t})] \, p_n\right) \cdot x_n + C\delta t^2 \left(\|p_n\|^2 + \|\nabla f(x_n)\|^2\right).$$

The assumption $0 < m < \nabla^2 f(x) \leq M$ together with the equalities $f(0) = 0$ and $\nabla f(0) = 0$ implies through Taylor expansions with integral remainder that $f(x) \geq m\|x\|^2/2$ and $\|\nabla f(x)\| \leq M\|x\|$. In view of these two inequalities, there exists $K \in \mathbf{R}_+$ such that, uniformly in $\varepsilon \in (0, 1/2]$,

$$\|p\|^2 + \|\nabla f(x)\|^2 \leq K\mathcal{W}(x, p, \xi).$$

Since $B_{\delta t} > C_{\delta t}^2/2$ and $A_{\delta t} > 0$, we have $A_{\delta t}\rho\|\xi_n\|^2 \leq \mathcal{W}(x_n, p_n, \xi_n)$ so that $\|\xi_n\| \leq \sqrt{\mathcal{W}(x_n, p_n, \xi_n)/(A_{\delta t}\rho)}$. Let us fix $R > 0$ and define $\mathscr{W}_R = A_{\delta t}\rho R^2$. Then, whenever $\mathcal{W}(x_n, p_n, \xi_n) \leq \mathscr{W}_R$ holds, we can deduce $\|\xi_n\| \leq R$ and therefore

$$\|I - \beta_{n,\delta t}\| = \|I - \mathrm{e}^{-(\gamma I + \Xi_n)\delta t}\| \leq 1 - \mathrm{e}^{-(\gamma+R)\delta t}.$$

We now conclude the proof by induction. Assume $\mathcal{W}(x_n, p_n, \xi_n) \leq \mathscr{W}_R$. Using the above bound on $\|I - \beta_{n,\delta t}\|$, we obtain

$$\mathcal{W}(x_{n+1}, p_{n+1}, \xi_{n+1}) \leq (1 + CK\delta t^2)\mathcal{W}(x_n, p_n, \xi_n) - \frac{\alpha\rho\delta t}{2}\|\xi_n\|^2 - \frac{\gamma\delta t}{2}\|p_n\|^2 - \varepsilon\delta t \, x_n \cdot \nabla f(x_n)$$
$$+ \varepsilon \left(2\delta t + 1 - \mathrm{e}^{-(\gamma+R)\delta t}\right) \|p_n\| \|x_n\|$$
$$\leq (1 + CK\delta t^2)\mathcal{W}(x_n, p_n, \xi_n) - \frac{\alpha\rho\delta t}{2}\|\xi_n\|^2$$
$$- \varepsilon\delta t \left(x_n \cdot \nabla f(x_n) - \eta \frac{|2\delta t + 1 - \mathrm{e}^{-(\gamma+R)\delta t}|}{\delta t}\|x_n\|^2\right)$$
$$- \delta t \left(\frac{\gamma}{2} - \frac{\varepsilon}{4\eta} \frac{|2\delta t + 1 - \mathrm{e}^{-(\gamma+R)\delta t}|}{\delta t}\right) \|p_n\|^2.$$

Note that we can then use the strong convexity inequality to write $-x_n \cdot \nabla f(x_n) \leq -f(x_n) - \frac{m}{2}\|x_n\|^2 \leq -m\|x_n\|^2$, upon which, the third term in the above inequality can be bounded as

$$-\varepsilon\delta t \left(x_n \cdot \nabla f(x_n) - \eta \frac{|2\delta t + 1 - \mathrm{e}^{-(\gamma+R)\delta t}|}{\delta t}\|x_n\|^2\right) \leq -\varepsilon\delta t \left(m - \eta \frac{|2\delta t + 1 - \mathrm{e}^{-(\gamma+R)\delta t}|}{\delta t}\right) \|x_n\|^2.$$

The constant $\eta$ then needs to be chosen sufficiently small so that $m - \eta \dfrac{|2\delta t + 1 - \mathrm{e}^{-(\gamma+R)\delta t}|}{\delta t} > 0$. Next, we must choose a sufficiently small $\varepsilon$ in order to ensure positivity of the term $\dfrac{\gamma}{2} - \dfrac{\varepsilon}{4\eta} \dfrac{|2\delta t + 1 - \mathrm{e}^{-(\gamma+R)\delta t}|}{\delta t} > 0$. We can then group terms of order $\delta t$, upon which we get a term of the form $O(\delta t)\left(\|x_n\|^2 + \|p_n\|^2 + \|\xi_n\|^2\right)$. From the definition of the Lyapunov function and upon completing the squares we can obtain an inequality of the form

$$\mathcal{W}(x_n, p_n, \xi_n) \leq C\left(\|x_n\|^2 + \|p_n\|^2 + \|\xi_n\|^2\right).$$

The above calculations imply that there exists $\kappa > 0$ such that

$$\mathcal{W}(x_{n+1}, p_{n+1}, \xi_{n+1}) \leq (1 - \kappa\delta t + CK\delta t^2)\mathcal{W}(x_n, p_n, \xi_n).$$

For a sufficiently small step-size $\delta t$, so that $0 < 1 - \kappa\delta t + CK\delta t^2 < \mathrm{e}^{-\kappa\delta t/2}$, the right hand side of the above inequality can be upper bounded by $\mathscr{W}_R$, which proves that the induction hypothesis also holds for $n + 1$. This allows us to conclude to the claimed exponential convergence result. $\qquad\square$

# I. CD Convergence Analysis

This section establishes the convergence properties of the proposed CD optimizer. We first prove that the continuous-time CD dynamics converge at an exponential rate to the global minimum. We then show that this exponential (geometric) rate is preserved in discrete time, provided the step size $\delta t$ is sufficiently small.

## I.1. Dynamical Properties of CD

In this section we study the dynamical properties of the system (7). We propose a Lyapunov function to prove exponential convergence of the dynamics under the assumption that $f$ is strongly convex.
A first result is that the dynamics is well posed on infinite time horizons under some conditions on $f$.

**Lemma I.1.** *Assume that $f$ is a smooth function, such that $f(x) \to \infty$ as $\|x\| \to +\infty$. Then, for any initial condition $(x_0, p_0) \in \mathbf{R}^d \times \mathbf{R}^d$, the solution of (7) is well defined for all times $t \geq 0$ and there exists $R \in \mathbf{R}_+$ such that $\|x(t)\| \leq R$ and $\|p(t)\| \leq R$, for any $t \geq 0$.*

*Proof.* Existence and uniqueness of a local in time solution of (7) follow from the Cauchy-Lipschitz theorem. To prove that the solution is global in time we define the following Lyapunov function

$$\mathcal{G}(x, p, \xi) = f(x) - f(x^*) + \frac{1}{2}\|p\|^2.$$

A simple computation shows that the function $\mathscr{G}(t) = \mathcal{G}(x(t), p(t), \xi(t))$ satisfies

$$\dot{\mathscr{G}}(t) = \nabla f(x(t)) \cdot \dot{x}(t) + p(t) \cdot \dot{p}(t)$$
$$= -\gamma\|p(t)\|^2 - cp(t) \cdot [p(t)]^3 \leq 0,$$

since $\gamma, c \geq 0$. This means that $\mathscr{G}(t) \leq \mathscr{G}(0)$, from which the result easily follows since $f - f(x^*)$ and $\|p\|^2$ are non-negative. $\qquad\square$

The next proposition, whose proof is immediate, characterizes the equilibria of the dynamics.

**Proposition I.2.** *The equilibria of the system (7) coincide with the physical equilibria, i.e.*

$$(\dot{x}, \dot{p}) = 0 \Leftrightarrow (p^*, \nabla f(x^*)) = 0.$$

## I.2. Exponential Convergence of the Continuous CD Dynamics

**Theorem 4.3.** *Consider a function $f \in C^2$. Assume that $\gamma, c > 0$ and that there exist $a, b > 0$ such that*

$$a\big[f(x)-f(x^*)\big] + b\|x - x^*\|^2$$
$$\leq (x - x^*) \cdot (\nabla f(x) - \nabla f(x^*)). \tag{11}$$

*Then, for any initial condition $(x_0, p_0) \in \mathbf{R}^d \times \mathbf{R}^d$, there exist $\kappa > 0$ and $C \in \mathbf{R}_+$ such that the solution of (7) satisfies*

$$f(x(t)) - f(x^*) + \|p(t)\|^2 \leq Ce^{-\kappa t}.$$

*Proof.* Similarly to the steps followed for the proof of Theorem 4.1, we consider the Lyapunov function

$$\mathcal{W}_\varepsilon(x, p) = f(x) - f(x^*) + \frac{1}{2}\|p\|^2 + \varepsilon(x - x^*) \cdot p + \varepsilon\|x - x^*\|^2, \tag{26}$$

for which the following upper bound can be derived (for $\varepsilon \in (0, 1/2]$):

$$\mathcal{W}_\varepsilon(x, p) \leq f(x) - f(x^*) + \frac{3}{4}\|p\|^2 + \frac{3\varepsilon}{2}\|x - x^*\|^2. \tag{27}$$

We also have the corresponding lower bound

$$\mathcal{W}_\varepsilon(x, p) \geq f(x) - f(x^*) + \frac{1}{4}\|p\|^2 + \frac{\varepsilon}{2}\|x - x^*\|^2.$$

Differentiating $\mathscr{W}_\varepsilon(t) = \mathcal{W}_\varepsilon(x(t), p(t))$ with respect to time, we obtain

$$
\begin{aligned}
\dot{\mathscr{W}}_\varepsilon(t) = & \nabla f(x(t)) \cdot \dot{x}(t) + p(t) \cdot \dot{p}(t) + \varepsilon \dot{x}(t) \cdot p(t) \\
& + \varepsilon(x(t) - x^*) \cdot \dot{p}(t) + 2\varepsilon(x(t) - x^*) \cdot \dot{x}(t) \\
= & -(\gamma - \varepsilon)\|p(t)\|^2 - \varepsilon(x(t) - x^*) \cdot \nabla f(x(t)) \\
& + \varepsilon(2 - \gamma)(x(t) - x^*) \cdot p(t) - c\varepsilon(x(t) - x^*) \cdot [p(t)]^3 - cp(t) \cdot [p(t)]^3 \\
\leq & -(\gamma - \varepsilon)\|p(t)\|^2 - \varepsilon(x(t) - x^*) \cdot \nabla f(x(t)) \\
& + \varepsilon|\gamma - 2|\left(\Delta\|x(t) - x^*\|^2 + \frac{1}{4\Delta}\|p(t)\|^2\right) + c\varepsilon\left(\eta\|x(t) - x^*\|^2 + \frac{R^4}{4\eta}\|p(t)\|^2\right),
\end{aligned}
$$

where we used a Cauchy–Schwarz inequality to bound the last two terms, as well as Lemma I.1 to bound the term $\|[p]^3\|^2$ as follows: $\|[p]^3\|^2 = \sum(p_i^3)^2 = \sum(p_i^2)^3 \leq (\sum p_i^2)^3 = (\|p\|^2)^3 \leq \|p\|^2 R^4$. We have also used the fact that $p \cdot p^3 \geq 0$.

In view of Condition (11), we can write

$$
\dot{\mathscr{W}}_\varepsilon(t) \leq -a\varepsilon(f(x(t)) - f(x^*)) - \left(\gamma - \varepsilon\left(1 + \frac{(\gamma - 2)^2}{b} + c^2\frac{R^4}{b}\right)\right)\|p(t)\|^2 - \frac{b\varepsilon}{2}\|x(t) - x^*\|^2,
$$

where we set $\Delta = \frac{b}{4|\gamma - 2|}$ and $\eta = \frac{b}{4c}$. This bound for $\dot{\mathscr{W}}_\varepsilon(t)$, combined with the upper bound in equation (27), yields

$$
\dot{\mathscr{W}}_\varepsilon(t) \leq -\min\left\{a\varepsilon, \frac{b}{3}, \frac{4}{3}\left(\gamma - \varepsilon\left(1 + \frac{(\gamma - 2)^2}{b} + c^2\frac{R^4}{b}\right)\right)\right\}\mathscr{W}_\varepsilon.
$$

For a sufficiently small $\varepsilon$ we can use Gronwall's inequality to conclude to an exponential convergence to 0 of $\mathscr{W}_\varepsilon(t)$ and hence convergence of $f(x(t)) - f(x^*) + \|p(t)\|^2$ thanks to the lower bound on the Lyaqpunov function. $\square$

## I.3. Discrete Convergence Analysis

We consider the following Euler discretization of the continuous CD equations (7), which is simpler to analyze than a splitting scheme:

$$
\begin{aligned}
p_{n+1} &= (1 - \gamma\delta t)p_n - c\delta t[p_n]^3 - \delta t\nabla f(x_n), \\
x_{n+1} &= x_n + \delta t p_n,
\end{aligned}
$$

where we assume that $\gamma\delta t$ is sufficiently small. The above discretization scheme coincides with Equations (12a)-(12b) and is restated here for completeness. We next consider a perturbation of the Lyapunov function (26), namely

$$
\mathcal{W}_{\delta t}(x_n, p_n) = f(x_n) + A_{\delta t}\frac{\|p_n\|^2}{2} + B_{\delta t}\|x_n\|^2 + C_{\delta t}x_n \cdot p_n, \tag{28}
$$

where $A_{\delta t} > 0$ and $B_{\delta t} > \frac{C_{\delta t}^2}{2A_{\delta t}}$.

**Theorem 4.4.** *Consider $f \in C^2$ and assume that there exist $m, M \in \mathbf{R}_+$ such that $m \leq \nabla^2 f(x) \leq M$ for any $x \in \mathbf{R}^d$. Fix $L > 0$. Then, for any initial condition $(x_0, p_0)$ such that $\|x_0\| + \|p_0\| \leq L$, there exist $\delta t^* > 0, r > 0$ and $C > 0$ for which, for any $n \geq 0$ and for any $\delta t \in (0, \delta t^*)$,*

$$
f(x_n) - f(x^*) + \|p_n\|^2 \leq Ce^{-\kappa n\delta t}.
$$

*Proof.* We begin the proof by upper bounding/expanding each of the terms in the Lyapunov function at step $n + 1$. Since $0 \leq \nabla^2 f(x) \leq M$, we have

$$
f(x_{n+1}) \leq f(x_n) + \delta t\nabla f(x_n) \cdot p_n + \frac{\delta t^2 M}{2}\|p_n\|^2.
$$

We assume that $\gamma \delta t \leq 1$. We can bound terms of the form $\|[p]^3\|$ similarly to the continuous case, provided there exists $R > 0$ such that $\|p\| < R$. Such a bound on $\|p\|$ can be derived by induction. Since $B_{\delta t} > C_{\delta t}^2/(2A_{\delta t})$ and $A_{\delta t} > 0$, the quadratic form in $(x, p)$ is positive definite and, together with $f(x) \geq \frac{m}{2}\|x\|^2$, this implies that there exists $c_0 > 0$ such that $c_0\|p_n\|^2 \leq c_0(\|x_n\|^2 + \|p_n\|^2) \leq \mathcal{W}(x_n, p_n)$. Hence, $\|p_n\| \leq \sqrt{\mathcal{W}(x_n, p_n)/c_0}$. Fix $R > 0$ and set $\mathscr{W}_R = c_0 R^2$. Then, whenever $\mathcal{W}(x_n, p_n) \leq \mathscr{W}_R$ holds, we can deduce $\|p_n\| \leq R$. Fix $L > 0$ such that $\mathcal{W}(x_0, p_0) \leq \mathscr{W}_R$ and proceed by induction assuming that $\mathcal{W}(x_n, p_n) \leq \mathscr{W}_R$. First,

$$
\begin{aligned}
\|p_{n+1}\|^2 =& \|(1-\gamma\delta t)p_n - c\delta t[p_n]^3 - \delta t\nabla f(x_n)\|^2 \\
=& \left\|(1-\gamma\delta t)p_n - c\delta t[p_n]^3\right\|^2 - 2\delta t\Big[(1-\gamma\delta t)p_n - c\delta t[p_n]^3\Big] \cdot \nabla f(x_n) + \|\nabla f(x_n)\|^2 \\
=& \|(1-\gamma\delta t)p_n\|^2 - 2c(1-\gamma\delta t)\delta t p_n \cdot [p_n]^3 + \|c\delta t[p_n]^3\|^2 \\
& - 2\Big[(1-\gamma\delta t)p_n - c\delta t[p_n]^3\Big]\delta t\nabla f(x_n) + \|\delta t\nabla f(x_n)\|^2 \\
\leq& (1-\gamma\delta t)^2\|p_n\|^2 + c^2\delta t^2 R^4\|p_n\|^2 \\
& - 2\delta t(1-\gamma\delta t)p_n \cdot \nabla f(x_n) + 2c\delta t^2[p_n]^3 \cdot \nabla f(x_n) + \delta t^2\|\nabla f(x_n)\|^2 \\
\leq& \|p_n\|^2 - 2\gamma\delta t\|p_n\|^2 + \gamma^2\delta t^2\|p_n\|^2 + c^2\delta t^2 R^4\|p_n\|^2 \\
& - 2\delta t(1-\gamma\delta t)p_n \cdot \nabla f(x_n) + 2c\delta t^2\Big(\frac{1}{4\theta}\|[p_n]^3\|^2 + \theta\|\nabla f(x_n)\|^2\Big) + \delta t^2\|\nabla f(x_n)\|^2 \\
\leq& \|p_n\|^2 - 2\gamma\delta t\|p_n\|^2 + \gamma^2\delta t^2\|p_n\|^2 + c^2\delta t^2 R^4\|p_n\|^2 \\
& - 2\delta t(1-\gamma\delta t)p_n \cdot \nabla f(x_n) + 2c\delta t^2\Big(\frac{R^4}{4\theta}\|p_n\|^2 + \theta\|\nabla f(x_n)\|^2\Big) + \delta t^2\|\nabla f(x_n)\|^2 \\
=& \|p_n\|^2 - 2\gamma\delta t\|p_n\|^2 + \gamma^2\delta t^2\|p_n\|^2 + c^2\delta t^2 R^4\|p_n\|^2 \\
& - 2\delta t p_n \cdot \nabla f(x_n) + 2\gamma\delta t^2 p_n \cdot \nabla f(x_n) + \frac{cR^4}{2\theta}\delta t^2\|p_n\|^2 + 2c\delta t^2\theta\|\nabla f(x_n)\|^2 + \delta t^2\|\nabla f(x_n)\|^2 \\
\leq& \|p_n\|^2 - 2\gamma\delta t\|p_n\|^2 + \gamma^2\delta t^2\|p_n\|^2 + c^2 R^4\delta t^2\|p_n\|^2 - 2\delta t p_n \cdot \nabla f(x_n) \\
& + 2\gamma\delta t^2\Big(\frac{1}{4\xi}\|p_n\|^2 + \xi\|\nabla f(x_n)\|^2\Big) + \frac{cR^4}{2\theta}\delta t^2\|p_n\|^2 + 2c\theta\delta t^2\|\nabla f(x_n)\|^2 + \delta t^2\|\nabla f(x_n)\|^2.
\end{aligned}
$$

Therefore

$$
\|p_{n+1}\|^2 \leq \|p_n\|^2 - 2\gamma\delta t\|p_n\|^2 - 2\delta t p_n \cdot \nabla f(x_n) + Q\delta t^2\Big(\|p_n\|^2 + \|\nabla f(x_n)\|^2\Big).
$$

We also have

$$
\|x_{n+1}\|^2 = \|x_n\|^2 + 2\delta t x_n \cdot p_n + \delta t^2\|p_n\|^2.
$$

Finally,

$$
\begin{aligned}
x_{n+1} \cdot p_{n+1} =& \Big(x_n + \delta t p_n\Big) \cdot \Big((1-\gamma\delta t)p_n - c\delta t[p_n]^3 - \delta t\nabla f(x_n)\Big) \\
=& (1-\gamma\delta t)x_n \cdot p_n - c\delta t x_n \cdot [p_n]^3 - \delta t x_n \cdot \nabla f(x_n) \\
& + (1-\gamma\delta t)\delta t p_n \cdot p_n - c\delta t^2 p_n \cdot [p_n]^3 - \delta t^2 p_n \cdot \nabla f(x_n) \\
\leq& x_n \cdot p_n - \gamma\delta t x_n \cdot p_n + c\delta t|x_n \cdot [p_n]^3| - \delta t x_n \cdot \nabla f(x_n) \\
& + \delta t\|p_n\|^2 - \gamma\delta t^2\|p_n\|^2 + \delta t^2\Big(\frac{1}{4\kappa}\|p_n\|^2 + \kappa\|\nabla f(x_n)\|^2\Big) \\
\leq& x_n \cdot p_n - \gamma\delta t x_n \cdot p_n + c\delta t|x_n \cdot [p_n]^3| - \delta t x_n \cdot \nabla f(x_n) \\
& + \delta t\|p_n\|^2 + \delta t^2\Big(\frac{1}{4\kappa}\|p_n\|^2 + \kappa\|\nabla f(x_n)\|^2\Big).
\end{aligned}
$$

We thus have

$$
\begin{aligned}
\mathcal{W}(x_{n+1}, p_{n+1}) \leq & f(x_n) + \delta t \nabla f(x_n) \cdot p_n \\
& + A_{\delta t} \frac{\|p_n\|^2}{2} - A_{\delta t}\gamma\delta t\|p_n\|^2 - A_{\delta t}\delta t \nabla f(x_n) \cdot p_n \\
& + B_{\delta t}\|x_n\|^2 + 2B_{\delta t}\delta t\|x_n\|\|p_n\| \\
& + C_{\delta t}x_n \cdot p_n + C_{\delta t}(\gamma + cR^2)\delta t\|x_n\|\|p_n\| - C_{\delta t}\delta t x_n \cdot \nabla f(x_n) + C_{\delta t}\delta t\|p_n\|^2 \\
& + Q\delta t^2 \Big(\|p_n\|^2 + \|\nabla f(x_n)\|^2\Big) \\
= & \mathcal{W}(x_n, p_n) + (1 - A_{\delta t})\delta t \nabla f(x_n) \cdot p_n \\
& - A_{\delta t}\gamma\delta t\|p_n\|^2 + 2B_{\delta t}\delta t\|x_n\|\|p_n\| \\
& + C_{\delta t}(\gamma + cR^2)\delta t\|x_n\|\|p_n\| - C_{\delta t}\delta t x_n \cdot \nabla f(x_n) + C_{\delta t}\delta t\|p_n\|^2 \\
& + Q\delta t^2 \Big(\|p_n\|^2 + \|\nabla f(x_n)\|^2\Big) \\
\leq & \mathcal{W}(x_n, p_n) - C_{\delta t}\delta t x_n \cdot \nabla f(x_n) + (1 - A_{\delta t})\delta t \nabla f(x_n) \cdot p_n \\
& - (A_{\delta t}\gamma - C_{\delta t})\delta t\|p_n\|^2 + \Big[2B_{\delta t} + C_{\delta t}(\gamma + cR^2)\Big]\delta t\|x_n\|\|p_n\| \\
& + Q\delta t^2 \Big(\|p_n\|^2 + \|\nabla f(x_n)\|^2\Big).
\end{aligned}
$$

In order to eliminate the term $\nabla f(x_n) \cdot p_n$, we set $A_{\delta t} = 1$, and we also demand $C_{\delta t} < \gamma$, upon which we have

$$
\begin{aligned}
\mathcal{W}(x_{n+1}, p_{n+1}) \leq & \mathcal{W}(x_n, p_n) - C_{\delta t}\delta t x_n \cdot \nabla f(x_n) - (\gamma - C_{\delta t})\delta t\|p_n\|^2 \\
& + \Big[2B_{\delta t} + C_{\delta t}(\gamma + cR^2)\Big]\delta t\|x_n\|\|p_n\| + Q\delta t^2 \Big(\|p_n\|^2 + \|\nabla f(x_n)\|^2\Big).
\end{aligned}
$$

Assuming that $x^* = 0$, $f(x^*) = 0$ (where $\nabla f(x^*) = 0$) and using the strong convexity and Lipschitz continuity assumption we have $f(x) \geq m\frac{\|x\|^2}{2}$ and $\|\nabla f(x)\|^2 \leq M^2\|x\|^2$. These inequalities imply that there exists $K \in \mathbf{R}_+$ such that

$$
\|p\|^2 + \|\nabla f(x)\|^2 \leq K\mathcal{W}(x, p)
$$

for $B_{\delta t} > C_{\delta t}^2/2$. This allows us to upper bound $\mathcal{W}(x_{n+1}, p_{n+1})$ as follows;

$$
\begin{aligned}
\mathcal{W}(x_{n+1}, p_{n+1}) \leq & (1 + QK\delta t^2)\mathcal{W}(x_n, p_n) - C_{\delta t}\delta t x_n \cdot \nabla f(x_n) \\
& - (\gamma - C_{\delta t})\delta t\|p_n\|^2 + \Big[2B_{\delta t} + C_{\delta t}(\gamma + cR^2)\Big]\delta t\|x_n\|\|p_n\|.
\end{aligned}
$$

Finally, using the strong convexity assumption and a Cauchy-Schwarz inequality we obtain

$$
\begin{aligned}
\mathcal{W}(x_{n+1}, p_{n+1}) \leq & (1 + QK\delta t^2)\mathcal{W}(x_n, p_n) - aC_{\delta t}\delta t f(x_n) \\
& - \left(C_{\delta t}b - \frac{\lambda^2}{2}\Big(2B_{\delta t} + C_{\delta t}(\gamma + cR^2)\Big)\right)\delta t\|x_n\|^2 \\
& - \left((\gamma - C_{\delta t}) - \Big(2B_{\delta t} + C_{\delta t}(\gamma + cR^2)\Big)\frac{1}{2\lambda^2}\right)\delta t\|p_n\|^2.
\end{aligned}
$$

Setting $B_{\delta t} = C_{\delta t} = \varepsilon$ and $\lambda^2 = \sqrt{\varepsilon}$, we obtain

$$\begin{aligned}
\mathcal{W}(x_{n+1}, p_{n+1}) \leq & (1 + QK\delta t^2)\mathcal{W}(x_n, p_n) - a\varepsilon\delta t f(x_n) \\
& - \left(\varepsilon b - \frac{\sqrt{\varepsilon}}{2}\left(2\varepsilon + \varepsilon(\gamma + cR^2)\right)\right)\delta t\|x_n\|^2 \\
& - \left((\gamma - \varepsilon) - \left(2\varepsilon + \varepsilon(\gamma + cR^2)\right)\frac{1}{2\sqrt{\varepsilon}}\right)\delta t\|p_n\|^2 \\
= & (1 + QK\delta t^2)\mathcal{W}(x_n, p_n) - a\varepsilon\delta t f(x_n) \\
& - \left(\varepsilon b - \varepsilon^{3/2}\left(1 + \frac{1}{2}\gamma + \frac{1}{2}cR^2\right)\right)\delta t\|x_n\|^2 \\
& - \left(\gamma - \varepsilon - \sqrt{\varepsilon}\left(1 + \frac{\gamma}{2} + \frac{cR^2}{2}\right)\right)\delta t\|p_n\|^2 \\
\leq & (1 - \nu\delta t + QK\delta t^2)\mathcal{W}(x_n, p_n).
\end{aligned}$$

Note we need to choose a sufficiently small $\varepsilon > 0$ in order for the coefficient of the $\|x_n\|^2$ and $\|p_n\|^2$ terms to remain negative. The above implies that there exists $\nu > 0$ such that

$$\mathcal{W}(x_{n+1}, p_{n+1}) \leq (1 - \nu\delta t + QK\delta t^2)\mathcal{W}(x_n, p_n).$$

Upon choosing a sufficiently small step size $\delta t$, so that $1 - \nu\delta t + QK\delta t^2 \leq 1$ we obtain $\mathcal{W}(x_{n+1}, p_{n+1}) \leq \mathcal{W}(x_n, p_n) \leq \mathscr{W}_R$. This closes the induction argument. The same result also allows us to conclude to the exponential convergence of the discrete dynamics. $\qquad\square$

### I.4. Convergence of Discretized CD Dynamics with Stochastic Gradients

We now show convergence of the discrete CD dynamics in the stochastic gradient setting. We include a stochastic-gradient version only for CD, since the noise shows up only in the gradient term and the proof is very similar to the deterministic case. We do not present a similar proof for iKFAD as the extra variable $\xi$ depends on the (noisy) momentum and feeds back into the momentum update, which in turn adds extra cross-terms to control. We leave the stochastic iKFAD proof for future work.

We use an Euler discretization similarly to (12a) - (12b), where the update of the momenta is performed based on the stochastic gradient.

$$\begin{aligned}
p_{n+1} &= (1 - \gamma\delta t)p_n - c\delta t[p_n]^3 - \delta t\hat{G}_\omega(x_n), \\
x_{n+1} &= x_n + \delta t p_n,
\end{aligned}$$

where we assume that $\gamma\delta t$ is sufficiently small. We denote by $\hat{G}_\omega(x)$ the stochastic gradient, where $\mathbb{E}[\hat{G}_\omega(x)] = \nabla f(x)$.

**Assumption I.3.** The variance of the stochastic gradients is uniformly bounded, that is there exists $D > 0$ such that, for all $x \in \mathbf{R}^d, \mathbb{E}\|\hat{G}_\omega(x) - \nabla f(x)\|^2 \leq D$.

**Proposition I.4.** *Assume that $f \in C^2$ and there exist $m, M \in \mathbf{R}_+$ such that $m \leq \nabla^2 f(x) \leq M$ for any $x \in \mathbf{R}^d$. Fix $L > 0$. Then, for any initial condition $(x_0, p_0)$ such that*

$$\|x_0\| + \|p_0\| \leq L,$$

*there exists $\delta t^* > 0, r > 0$ and $C > 0$ for which*

$$\forall n \geq 0 \text{ and } \forall \delta t \in (0, \delta t^*), \quad \mathbb{E}[f(x_n) - f(x^*)] + \mathbb{E}\|p_n\|^2 \leq Ce^{-\kappa n\delta t}.$$

We sketch the proof of this result. We again consider the Lyapunov function (28)

$$\mathcal{W}(x_n, p_n) = f(x_n) + A_{\delta t}\frac{\|p_n\|^2}{2} + B_{\delta t}\|x_n\|^2 + C_{\delta t}x_n \cdot p_n.$$

We start by upper bounding the expectation of each of the terms in the discrete Lyapunov function. Let us emphasize that the expectations below are conditional on $x_n, p_n$. We follow the same procedure as in section I.3. As we have assumed that $0 < m < \nabla^2 f(x) \leq M$, we have

$$f(x_{n+1}) \leq f(x_n) + \delta t \nabla f(x_n) \cdot p_n + \frac{\delta t^2 M}{2} \|p_n\|^2,$$

so that

$$\mathbb{E}[f(x_{n+1})] \leq f(x_n) + \delta t \nabla f(x_n) \cdot p_n + \frac{\delta t^2 M}{2} \|p_n\|^2.$$

Next, we can bound the expectation of the squared norm of the momenta (see the proof of Theorem 4.4, in Appendix I.3) as:

$$\|p_{n+1}\|^2 \leq \|p_n\|^2 - 2\gamma \delta t \|p_n\|^2 - 2\delta t p_n \cdot \hat{G}_\omega(x_n) + C\delta t^2 \Big( \|p_n\|^2 + \|\hat{G}_\omega(x_n)\|^2 \Big),$$

so that

$$\begin{aligned}
\mathbb{E}\left[ \|\hat{G}_\omega(x_n)\|^2 \right] &= \mathbb{E}\|\nabla f(x_n) + (\hat{G}_\omega(x_n) - \nabla f(x_n))\|^2 \\
&= \|\nabla f(x_n)\|^2 + 2\nabla f(x_n) \cdot \mathbb{E}\left[ \hat{G}_\omega(x_n) - \nabla f(x_n) \right] + \mathbb{E}\|\hat{G}_\omega(x_n) - \nabla f(x_n)\|^2 \\
&\leq \|\nabla f(x_n)\|^2 + D,
\end{aligned}$$

since $\mathbb{E}\left[ \hat{G}_\omega(x_n) - \nabla f(x_n) \right] = \mathbb{E}\left[ \hat{G}_\omega(x_n) \right] - \nabla f(x_n) = 0$. Therefore,

$$\begin{aligned}
\mathbb{E}\|p_{n+1}\|^2 &\leq \|p_n\|^2 - 2\gamma \delta t \|p_n\|^2 - 2\delta t p_n \cdot \nabla f(x_n) + C\delta t^2 \Big( \|p_n\|^2 + \mathbb{E}\|\hat{G}_\omega(x_n)\|^2 \Big) \\
&\leq \|p_n\|^2 - 2\gamma \delta t \|p_n\|^2 - 2\delta t p_n \cdot \nabla f(x_n) + C\delta t^2 \Big( \|p_n\|^2 + \|\nabla f(x_n)\|^2 + D \Big).
\end{aligned}$$

Next, we have:

$$\mathbb{E}\|x_{n+1}\|^2 = \|x_n\|^2 + 2\delta t x_n \cdot p_n + \delta t^2 \|p_n\|^2.$$

Finally

$$\begin{aligned}
x_{n+1} \cdot p_{n+1} \leq{}& x_n \cdot p_n + (\gamma + cR^2)\delta t \|x_n\|\|p_n\| - \delta t x_n \cdot \hat{G}_\omega(x_n) + \delta t \|p_n\|^2 \\
&+ C\delta t^2 \Big( \|p_n\|^2 + \|\hat{G}_\omega(x_n)\|^2 \Big),
\end{aligned}$$

so that

$$\begin{aligned}
\mathbb{E}[x_{n+1} \cdot p_{n+1}] \leq{}& x_n \cdot p_n + (\gamma + cR^2)\delta t \|x_n\|\|p_n\| - \delta t x_n \cdot \nabla f(x_n) + \delta t \|p_n\|^2 \\
&+ C\delta t^2 \Big( \|p_n\|^2 + \|\nabla f(x_n)\|^2 + D \Big).
\end{aligned}$$

Having derived the above upper bounds, convergence can be shown following the same steps as in Section I.3.

