# OpenReview forum: "Adaptive Momentum and Nonlinear Damping for Neural Network Training"
_ICML.cc/2026/Conference — ICML 2026 regular_

### Official Review · Reviewer_ySqV · 2026-03-04

**Soundness:** 2
**Presentation:** 2
**Significance:** 2
**Originality:** 4
**Overall Recommendation:** 2
**Confidence:** 4

**Summary:**

It is argued that SGD with momentum underperforms Adam in many tasks
because of lack of adaptivity and coordinate-wise heterogeneity. Employing physical intuition, the authors extend SGD with momentum by including coordinate-wise adaptive friction coefficients (in continuous-time formulations, which are then discretized as necessary), leading to an optimizer called Individual Kinetic Friction Adaptive Descent (iKFAD). Using a near-equilibrium analysis, they find that such adaptive friction is linked to cubic damping, leading to another optimizer that largely matches the iKFAD performance, called Cubically Damped momentum SGD (CD). They also introduce cubic damping to Adam, obtaining a third optimizer Cubically damped Adam (CADAM). They provide exponential convergence guarantees for iKFAD and CD. In small vision and language experiments, they demonstrate that their extensions of momentum SGD match or outperform Adam, showcasing the correctness of the physical intuitions.

**Compliance With Llm Reviewing Policy:**

Affirmed.

**Final Justification:**

I maintain my score for the reasons listed in weaknesses and rebuttal acknowledgment: it is a real improvement that we worked with the authors to reduce overclaiming but the evidence at this point is not convincing enough (not strong enough baselines, not strong enough tasks) to justify the claim the authors make in the tl&dr: “We improve momentum SGD ... resulting in performance gains competitive with Adam on transformer tasks.” (Please note that I'm not at all expecting state-of-the-art performance even on NanoGPT-scale tasks: that would be a much higher bar.)

**Key Questions For Authors:**

Responses to neither of the questions are likely to increase my score but they might be important to clarify anyway. (Please refer to weaknesses for suggestions on what would warrant increasing my score.)

1. In the conclusion, it is claimed that your proposed methods are more robust to learning rate choice than Adam. Where is evidence of this? In Figure 7, you compare to SGD.

2. In the conclusion, it is claimed that your proposed methods generalize better than Adam. Did I miss it or is there no careful comparison of generalization specifically?

3. Do random seeds parametrize model initialization only, or the data batches as well (or is this task-dependent)?

**Limitations:**

Yes

**Strengths And Weaknesses:**

**Strengths**:
- The paper is clearly written (apart from excessive wording) and well-proofread.
- Theoretical claims are supported by rigorous proofs.
- Good physical motivations are included, and this paper connects to literature on vibration and acoustics.
- An equal hyperparameter tuning budget was allocated to baseline and proposed methods, which is rare and nice.
- Best test loss tables include standard deviations over 10 runs, and some figures have error bars.

**Weaknesses**:
- One major weakness is that the tasks are extremely small. I understand that it can be difficult to obtain compute for research, but it is (unfortunately) still important to present convincing evidence when proposing optimization algorithms.
The largest GPT-2 model used in the paper has 45M parameters. This is 2.6 times smaller than GPT-2-Small. The best loss achieved by this model on OpenWebText is 3.691. I would say this is too high a loss to make conclusions. In addition, I am not sure what is the parameter count of "GPT-2-Nano" in the paper. The reported figure is 0.85M. If that is not a typo, this experiment should be discarded. The compute budget of 80 trials for Bayesian hyperparameter tuning is also very small, and it is not clear to me how to make conclusions from this. For example, as a very important part, it would be beneficial to see at least some evidence that the learning rates are not wildly suboptimal (apart from the number of trials used in the Bayesian search), and that tuned Adam outperforms baseline Adam :)
- Relatedly to the previous point, weight decay / learning rate warmup and decay (for language tasks) and regularization / data augmentation (for vision tasks) are not used. This, in combination with task size, raises serious concerns about whether the results are relevant to the real practice of deep learning. To clarify, I understand that tiny experiments can be enough to capture some particular phenomena, but to accept a paper proposing new optimizers, I think the standard should be higher than good speculations, convergence proofs and toy tasks. I would suggest taking a baseline according to modern practices, like NanoGPT or (better) Llama model of 120M-350M parameters trained on OpenWebText or Fineweb to achieve the loss in the ballpark of 3.1-3.3. One such experiment with a reasonable baseline, reproducing the conclusions of the paper, would convince me that these conclusions are likely to be relevant to practical settings, and the new optimizers deserve attention at the conference.
- Adam is within error bars from the proposed methods in some experiments and it should be mentioned more visibly, and some claims about the good performance of the proposed methods might not be well-supported (correct me if I am wrong; please see questions).
- As a minor point, some explanations seem pretty long-winded and could be made more concise. For example, consider the paragraph starting with "Even though momentum-based methods..." on page 2. It begins with arguing that it is important to tune the momentum parameter (not sure why it is necessary to explain this), and concludes with repeating a point already made above that we need to consider separate parameters separately. I would try to avoid redundancy and repetition. Perhaps, reducing the intuitive speculations about the loss landscape, or at least making them shorter and to the point, would be better. As further illustrations, I suspect that the text *Note that while Equation (6b) also includes a standard fixed friction term apart from the new adaptive friction mechanism, we find that in practice we can often set $\gamma \approx 0$ and still maintain performance. The existence of a fixed friction term however is helpful in obtaining theoretical convergence guarantees* can be shortened in half, and the sentence *In GPT2-Nano, iKFAD exhibits accelerated convergence relative to Adam, while CD converges more gradually, ultimately matching the other optimizers toward the end of training* is overly convoluted (consider *iKFAD converges faster than Adam, while CD is slower mid-training but ultimately reaches comparable performance*). Also, the last explanation is not very necessary because it is not reliable (at least the iKFAD vs. Adam difference is tiny).

---

> ### Author Rebuttal · Authors · 2026-03-31
>
> Response to "...the tasks are extremely small..." and  "...weight decay / learning rate warmup and [other devices] are not used..."
>
> While we acknowledge that GPT2-XS (45M) is smaller than state-of-the-art LLMs, our goal was to investigate whether nonlinear damping can bridge the fundamental Adam-mSGD performance gap in Transformers.
> - **Consistency Across Scales:** We observe identical patterns across scales ranging from NanoGPT (85K parameters) to GPT2-XS (45M). In both cases, iKFAD and CD consistently overcome the optimization instabilities that cause mSGD to fail in high-curvature Transformer landscapes.
> - **Resource Constraints:** Experimenting with models >120M parameters is unrealistic within the short rebuttal period due to our available compute. However, our results show consistent trends at multiple scales. Notably, other recent optimiser studies have utilized comparable scales to isolate specific phenomena, and we believe our current scope provides a statistically significant baseline for these new dynamics. (Note: We confirm the GPT2-Nano parameter count is 85K; the 0.85M in the text was a typo ).  We will incorporate larger scale experimental results in the camera-ready version.  Our decision to omit learning rate (LR) schedules and complex regularization was a deliberate scientific choice to isolate the intrinsic stability of the optimiser’s dynamics.
> - **Controlled Comparison:** Introducing LR warmup and decay can "mask" an optimiser's tendency to oscillate or diverge. By keeping the learning rate constant, we demonstrate that iKFAD and CD possess inherent stabilization mechanisms (analogous to physical "emergency brakes") that maintain progress where mSGD typically becomes unstable.
> - **Clarification on Practices:** Weight decay was included in our experiments, though it was not a tuned hyperparameter to maintain parity across search budgets. For vision tasks (Tiny ViT and ResNet-18), we did employ standard data augmentation to reflect practical settings. Specifically, our training pipeline included RandomCrop (32x32 with padding of 4) and RandomHorizontalFlip.
>
> Response to "Adam is within error bars from the proposed methods...":
> Please see answers to questions, below.
>
> Response to "...some explanations...could be more concise...":
> We appreciate the feedback on the manuscript's flow. We will significantly condense the intuitive descriptions of the loss landscape (Page 2) and shorten the technical discussion regarding fixed versus adaptive friction terms. We will also adopt the reviewer's suggested phrasing for the GPT2-Nano convergence summary to improve readability.
>
> Responses to key questions:
> 1) We thank the reviewer for pointing out this inconsistency in the conclusion claims. What we meant to say was that our methods are more robust with respect to the choice of step-size and adaptive friction coefficient $\gamma$ than mSGD as demonstrated in Figure 7. However, while we do not perform a step-size sensitivity analysis for Adam, we have empirically observed from our experiments that optimally tuned step-sizes for CD and iKFAD tend to be larger than optimal step-sizes for both Adam. We will revise the phrasing in the conclusion to more accurately reflect our results.
> 2) From our experiments, we observe that, in some cases, iKFAD and CD achieve lower test loss than Adam. However, we do not include a study on the difference in generalisation performance between our methods and Adam. We will therefore revise the conclusion to avoid overstating this point and state only that our methods often achieve competitive or better test loss than Adam.
> 3) For all experiments, random seeds parameterized both the model initialization and the data batch shuffling. This ensures that the averaged curves in Figure 4 account for both architectural and data-driven stochasticity.

---

> > ### Author Rebuttal · Reviewer_ySqV · 2026-04-03
> >
> > I appreciate the authors' responses! Some updates in the writing improve presentation, but the core concern remains. To accept a paper proposing new methods (or justify that a particular modification can close the gap between SGD and Adam), I think the evidence should be richer than convergence proofs on a strongly convex objective and toy tasks with potentially undertuned baselines.

---

> > > ### Author Response · Authors · 2026-04-07
> > >
> > > We thank the reviewer for acknowledging the improvements. Our intention was not to claim that the Adam-mSGD gap is resolved in general through the use of our optimisers, but rather, that the proposed methods substantially reduce this gap on the specific benchmarks we study. We will revise the wording throughout the paper to make this clear.
> > >
> > > The role of the convergence analysis in the paper is to provide a clean setting in which the proposed damping mechanisms can be understood and where exponential convergence for iKFAD and CD can be shown. We perform this analysis in both a continuous and a discrete-time setting under a strong convexity-type condition. We note that convexity is a standard assumption made in the theoretical analysis of optimisers, even in highly non-convex machine learning settings.
> > >
> > > Regarding our experiments and numerics, the lower-dimensional, non-machine-learning examples are meant only as illustrations of the qualitative behaviour of the dynamics (although note that quantitative improvement was visible in the anisotropic quadratic study). In addition to these results, the paper includes experiments on modern deep-learning benchmarks, namely TinyViT on CIFAR-10, DistilBERT on SST-2 and QNLI, and GPT2 models on Shakespeare and OpenWebText. Across these settings, we compare against Adam and mSGD, average results over 10 random seeds, and use the same hyperparameter-search budget for all optimizers. Our tuning process, in fact, gives an advantage to Adam and mSGD over iKFAD and CD respectively, as the latter two optimisers have more hyperparameters, but are tuned using the same number of trials as the baseline optimisers.
> > > Regarding the reviewer’s point about computational budget during hyperparameter tuning, we additionally conducted a 500-trial sweep for Nano-GPT(number of parameters 0.85 M) in order to assess the sensitivity of the hyperparameter tuning process with respect to the search budget and did not observe significant changes in the  resulting losses and optimiser performance, compared to those obtained using 80 trials. This suggests that our 80-trial budget is sufficient for a fair comparison across methods.
> > >
> > > Regarding the OpenWebText experiment, our test losses were higher than standard losses on this benchmark, due to the fact that we used a downsized model. However, we believe that despite not having achieved state-of-the-art performance on this dataset, the experiment remains qualitatively informative and useful for comparing optimiser behaviour under identical conditions.
> > >
> > > Finally, we note that in our experiments CD (along with iKFAD) achieves performance comparable to Adam, while retaining a simple momentum-based structure with the same number of hyperparameters as mSGD (when γ=0), suggesting that nonlinear damping alone can significantly improve optimization behavior in transformer-based machine learning problems.
> > >
> > > We appreciate the reviewer’s feedback on our experiments and promise to also provide results on larger transformer architectures in time for the camera-ready deadline.

---

### Official Review · Reviewer_TQAe · 2026-03-09

**Soundness:** 3
**Presentation:** 3
**Significance:** 2
**Originality:** 2
**Overall Recommendation:** 4
**Confidence:** 4

**Summary:**

Authors consider parameter evolution in an analytic framework inspired by classical mechanics. Specifically, they examine the effect of dissipative (friction-like) terms in the differential equations governing particle motion. Their aim is to obtain a formulation of momentum adapted to local curvature to both support efficient exploration and suppress nefarious oscillations. Building on Friction-Adaptive Descent (FAD) by Karoni et al, they investigate 3 alternative formulations of an adaptive dissipative term.

First, Individual Kinetic Friction Adaptive Descent (iKFAD) uses an adaptive friction coefficient for which the time derivative increases with the momentum-squared and otherwise has a self-dissipating term. The effect increases damping for parameters with a recent history of high energy (momentum-squared) states.

Second, Cubically Damped mSGD (CD) is a simplified formulation inspired by the dynamics of iKFAD. Since the dissipation in iKFAD effectively multiplies the recent momentum-squared by the current momentum, CD simply expresses these mechanics directly in a cubic damping term.

Third, Cubically Damped Adam (CADAM). This formulation effectively replaces the momentum EMA in Adam with the CD formulation. Otherwise, adaptive scaling by the EMA of the second moment remains the same.

While technically sound, the contribution appears incremental and lacks direct comparison to the most relevant baseline.

**Compliance With Llm Reviewing Policy:**

Affirmed.

**Final Justification:**

The authors addressed my remaining concern by comparing FAD baselines and showing a notable improvement with iKFAD on both TinyViT and GPT2-Nano.

**Key Questions For Authors:**

Both Theorems 4.1 and 4.3 use the assumption: a[ f(x) - f(x^*) ] + b || x - x^* ||^2 \leq (x - x^*)^T (\grad f(x) - \grad f(x^*)). Is it fair to say this is a stronger condition than strong convexity: b || x - x^* ||^2 \leq (x - x^*)^T (\grad f(x) - \grad f(x^*))? If so, why is the additional term needed? Does your convergence proof provide insight into what may drive failure modes?

What use case is CADAM intended to address? I see that test results are included in the appendix and that, while competitive in a couple experiments, it never acquires a notable lead over the alternatives.

**Limitations:**

yes

**Strengths And Weaknesses:**

# Strengths

They derive the equivalence to a discretized solver using symplectic Euler (semi-implicit Euler).  Aside from parameters and gradients, the number of additional parameter-length states required by iKFAD and CADAM is 2, the same as Adam. Likewise CD requires 1 extra, just as mSGD. They also present convergence proofs for iKFAD and CD.

Their tests showing the impact of dropping the linear damping coefficient, \gamma=0, show that it often has only a minor impact. For DistilBert, dropping linear damping performs better. At a minimum, this can mitigate the challenge of hyper-parameter search by providing a simple starting value.

# Weaknesses

This work appears to be an incremental improvement on FAD methodology, but the paper does not include direct comparisons to FAD in the main experimental section. Since the proposed methods appear to be variants or simplifications of that approach, it is difficult to assess whether the improvements arise from the new formulation or are already captured by the prior method.

It looks like iKFAD requires a total of 4 hyper-parameters. In addition to the learning-rate, the dynamical formulation in (6) requires \alpha, \gamma, \rho. While Adam has 3: learning, \beta1, and \beta2, they also come with robust default values. After my first reading, it was not clear to me how much effort was involved in tuning these additional hyper-parameters or if there are reasonable default values that could serve as a baseline.

The training experiment caption in Figure 4 references iKFAD, CD, CADAM, Adam, and mSGD, but descent curves for CADAM are not included.

Figure 5 only examines the effect of removing linear damping from iKFAD and CD. This experiment would apply to CADAM as well.

---

> ### Author Rebuttal · Authors · 2026-03-31
>
> Response to ``This work appears to be an incremental improvement on FAD methodology...'':
>
> We thank the reviewer for the opportunity to clarify our contribution relative to Friction Adaptive Descent (FAD). While our work is inspired by the FAD framework, it introduces three fundamental advances:
> - **Coordinate-wise Adaptation:** FAD utilizes a single friction coefficient for all parameters. In contrast, iKFAD has per-parameter adaptive friction. This is a critical distinction for deep learning; modern architectures like transformers exhibit extreme Hessian heterogeneity, where different layers require vastly different damping scales.
> - **New Methodology (CD):** Cubically Damped mSGD (CD) is a novel formulation that was only briefly introduced in the original FAD work. It offers a significant reduction in memory ($2N$ states) compared to FAD/iKFAD ($3N$ states) while matching Adam’s performance on key benchmarks.
> - **Application Domain:** FAD was originally developed for molecular dynamics and simple optimization. We provide the first rigorous application and theoretical analysis of these principles in the context of neural network training, specifically addressing the "Adam-mSGD gap".
>
> We also provide theoretical analysis for both iKFAD and CD (in continuous and discrete-time settings).
>
> Response to ``It looks like iKFAD requires a total of 4 hyper-parameters..baseline'':
>
> In its original formulation, iKFAD has four hyperparameters ($\delta t$, $\alpha$, $\gamma$, $\rho$). However, in practice we have observed that the linear damping coefficient can be set to zero  with no significant effect on performance (see Fig. 5). This reduces the number of hyperparameters for iKFAD to three, which is the same number of hyperparameters as Adam.
>
> We agree that the paper should be clear about tuning effort. The methods used  - including Adam and mSGD - were tuned under the same Optuna budget of 80 trials per experiment, in practice Adam hyperparameters are not usually tuned and the default values are used for $β_1$ and $β_2$ by most practitioners. While we cannot yet suggest fixed default values $\alpha$ and $\rho$ for iKFAD,  we observed that the quantity $1/(\alpha \rho)$, seems to depend strongly on the scale of the momenta and hence the gradient scale for each problem, suggesting that gradient/momenta normalization could  improve reliability.  We have made the decision to leave this technical matter for future work.
>
> Response to "The training experiment caption in Figure 4 references iKFAD, CD, CADAM, Adam, and mSGD...", and "Figure 5 only examines the effect of removing linear damping ...".
>
> We thank the reviewer for spotting these presentation issues. We will amend the label in Figure 4.  Cadam did not seem to offer significant benefits over Adam.
>
> Responses to Key Questions:
> 1) No, our assumption:
> $$a[ f(x) - f(x^{\*}) ] + b \| x - x^{\*} \|^2 \leq (x - x^{\*})^T (\nabla f(x) - \nabla f(x^{\*})) \tag{inequality A}$$
> is not stronger than standard $\mu$-strong convexity. A standard consequence of $\mu$-strong convexity is the following inequality where $x^{\*}$ is the minimiser:
> $$f(x) - f(x^{\*})  + \frac{\mu}{2} \|x - x^{\*} \|^2 \leq (x - x^{\*})^T (\nabla f(x) - \nabla f(x^{\*})) \tag{inequality B }$$
> while another - weaker consequence - of $\mu$-strong convexity is
> $$\mu \| x - x^{\*} \|^2 \leq (x - x^{\*})^T (\nabla f(x) - \nabla f(x^{\*})) \tag{inequality C}$$
> Strong convexity also gives
> $$ f(x) - f(x^{\*})   \geq (\mu/2) \| x - x^{\*} \|^2  \tag{inequality D}$$
> and combining (inequality D) and (inequality B) yields (inequality C).
> If $f$ is $μ$-strongly convex, then our assumption (A) holds automatically for the specific pair  $a=1$ and $b=\mu/2$. Standard $\mu$-strong convexity is more restrictive, since it implies the inequality for the specific choice $a=1$ and $b=\mu/2$, whereas our assumption only requires that some positive constants $a, b$ exist. Our assumption is a weaker condition.  We could have used the standard inequality (B) for the proof.  Our statement (A) using $a$ and $b$ is simply more general.
>
> Finally, the reason we do not use inequality (C) is that we want to control both $f(x)-f(x^{\*})$ and $\|x-x^{\*}\|^2$. Inequality (C) only allows control of $\|x-x^{\*}\|$, whereas condition (A) allows us to replace terms of the form   $(x - x^{\*})^T (\nabla f(x) - \nabla f(x^{\*}))$ by a combination of  $f(x)-f(x^{\*})$ and $\|x-x^{\*}\|^2$. This allows us to deduce the desired convergence in $f(x)$ and $x$.
>
> 2)  CADAM was developed to investigate the interplay between adaptive learning rates (Adam) and nonlinear damping (CD). The fact that CADAM does not significantly outperform Adam is a key scientific finding: it indicates that nonlinear damping and adaptive learning rates serve a similar stabilizing purpose. This discovery highlights that the primary benefit of iKFAD and CD is providing Adam-level stability to mSGD-style updates without the need for explicit gradient rescaling.

---

> > ### Author Rebuttal · Reviewer_TQAe · 2026-04-01
> >
> > I appreciate the clarification regarding the theoretical assumptions and the distinction from FAD. However, my central concern remains: the proposed methods are presented as advances over FAD, yet no direct experimental comparison to FAD is included. Could you clarify why this baseline was omitted, and whether such a comparison could be added?
> >
> > Regarding Figure 4: to clarify, the issue is not a mislabeled caption. CADAM is a proposed method and its training curves should be shown alongside the others, regardless of whether it outperforms them. Omitting results that are less favorable undermines confidence in the experimental presentation.

---

> > > ### Author Response · Authors · 2026-04-07
> > >
> > > We thank the reviewer for their constructive follow-up and for highlighting these points. The reviewer’s question about FAD performance has prompted us to do a more extensive comparison between KFAD and iKFAD on TinyViT and NanoGPT, which confirmed what our preliminary runs had hinted at; using KFAD (the original FAD formulation with a single adaptive friction coefficient), leads to significantly poorer performance than iKFAD and CD, which demonstrates that individual treatment of different parameters can be crucial in large machine learning tasks, especially those involving transformer-based architectures.
> > >
> > > We include a table below (see also as Table 1 in main paper), which summarizes the best test losses (mean and standard deviation, averaged over ten runs) for each of the two experiments, which also includes the new KFAD results (last column). The KFAD optimal hyperparameters were obtained with the same Bayesian search sweep budget, using the same hyper parameter ranges as iKFAD. Note that for Tiny-ViT specifically, while the reported best accuracy for KFAD (and mSGD) is not dramatically different to the best iKFAD loss, iKFAD reaches this loss value much faster (see for example iKFAD vs mSGD performance in the Tiny-ViT experiment in the first column of Figure 4) which leads to a significant gain in terms of computational cost. This will be more clearly illustrated through the full training and test loss curves, which we will include in the paper.
> > >
> > > Model (Dataset)       |     Adam        |       mSGD     |         CD       |      iKFAD     |                 KFAD
> > >
> > > TinyViT (CIFAR-10) | 0.613±0.018  | 0.647 ±0.012 | 0.619±0.016 | 0.612±0.014  | 0.6424±0.014
> > >
> > > GPT2-Nano (SPC)   | 1.647±0.010   | 1.784 ±0.011 | 1.664±0.008  | 1.641±0.006 | 1.7363±0.018
> > >
> > > Table 1: Best Test Loss
> > >
> > > These new results show that the original FAD method is closer in performance to mSGD, while our newly proposed methods iKFAD and CD can reach Adam-level performance on machine learning tasks where vanilla mSGD typically struggles.
> > > We would like to stress again that we do not view this work as a trivial extension of FAD  methodology. In this paper, apart from building on FAD to introduce iKFAD, which as just shown, has the potential to greatly outperform KFAD (as well as mSGD) on transformer-based architectures, we also study CD and CADAM, which have never been studied in an ML or non-ML setting before. We provide continuous and discrete convergence proofs for iKFAD and CD and have run a series of numerics to demonstrate the potential of iKFAD, CD and CADAM on modern machine learning problems. Furthermore, the potential of FAD-based methods for  machine learning problems had never been demonstrated, prior to this work.
> > > We agree that direct comparison to the original FAD method, would enhance the paper and strengthen our motivation for introducing individual adaptive frictions; we have, as mentioned, performed this comparison after the reviewer’s suggestion and will include it in the camera-ready version.
> > >
> > > Regarding Figure 4 and CADAM, our intention was not to hide weaker results, but to keep the main presentation focused on the two optimizers (out of the ones we proposed) that performed best, while placing the results of our less competitive optimiser, CADAM, in the appendix. We respectfully disagree with the referee on this point and still prefer the placement of these secondary results in supplement. We are willing to characterize these results more comprehensively with an expanded comment in the main manuscript body.

---

### Official Review · Reviewer_FTAt · 2026-03-12

**Soundness:** 3
**Presentation:** 3
**Significance:** 3
**Originality:** 3
**Overall Recommendation:** 5
**Confidence:** 3

**Summary:**

This paper introduces an optimization algorithm for training neural networks, motivated by a continuous-time dynamical systems perspective on momentum methods. The authors use a continuous-time formulation to introduce individual, adaptive momentum coefficients regulated by the kinetic energy of each model parameter. The two related optimizers that incorporate nonlinear damping mechanisms to improve stability and convergence are proposed. Exerimental exmaples verify the effectiveness of the proposed method.

**Compliance With Llm Reviewing Policy:**

Affirmed.

**Key Questions For Authors:**

1. The organization can be improved. For example, Section 3 normally should be placed before Section 2. Besides, the connections between different sections are not satisfactory.
2. In the verification part, the applied benchmark models are all under 100M. How well would the proposed iKFAD and CD perform when training a large-scale model?
3. In the theoretical analysis, the leap from the continuous-time proofs to the discrete-time algorithms may require the assumption of a sufficiently small step size. If this assumption does not hold, will the method still converge?
4. Is it possible to extend the optimization method to some other applications, such as in other fields, not only for NN training?

**Limitations:**

1. The applications of the proposed method seem to be limited since there may be some conditions.
2. The analysis and proofs are based on the continuous-time system, while in practice, the algorithms run in a discrete-time way.

**Strengths And Weaknesses:**

Strengths:
1. The proposed Individual Kinetic Friction Adaptive Descent method and Cubically Damped mSGD are the key contributions of this work.
2. These optimizers are interesting and the convergence analysis of the two methods is provided, which offers solid theoretical support.
3. Different datasets verify the advantages of the proposed method.

Weaknesses:
1. The organization can be improved. For example, Section 3 normally should be placed before Section 2. Besides, the connections between different sections are not satisfactory.
2. In the verification part, the applied benchmark models are all under 100M. How well would the proposed iKFAD and CD perform when training a large-scale model?
3. In the theoretical analysis, the leap from the continuous-time proofs to the discrete-time algorithms may require the assumption of a sufficiently small step size. If this assumption does not hold, will the method still converge?

---

> ### Author Rebuttal · Authors · 2026-03-30
>
> Responses to Key Questions:
> 1)  We thank the reviewer for this suggestion. We will move the related work section before the methodology section and improve transitions between sections. We will also refine the transitions between the continuous-time Hamiltonian motivation and the practical discrete-time implementations to ensure a more cohesive narrative.
> 2) While our current study focused on models under 100M parameters due to computational constraints, the underlying mechanics of iKFAD and CD are coordinate-wise and model-agnostic. Because the damping mechanism regulates the kinetic energy of each parameter individually, it accounts for the heterogeneity of modern loss landscapes regardless of total parameter count. Empirically, we observed that as we scaled from NanoGPT (0.85M) to GPT2-XS (45M), the optimal scale for the hyperparameters $\rho$ (iKFAD) and $c$ (CD) shifted predictably— $\rho$ decreased and $c$ increased—reflecting the evolving gradient scales. This trend suggests that our methods are well-suited for larger scales, requiring only standard hyperparameter tuning. We will provide results on larger transformer architectures in time for the camera-ready deadline.
> 3) The convergence proof in the discrete setting does require a sufficiently small step-size assumption. This ensures that the multiplier in the final inequality
> $$W_{n+1} \leq (1− \kappa \delta t  + C \delta t^2)W_n$$
> lies within the interval $(0,1)$, so the Lyapunov function still contracts at each step. If this assumption does not hold, our discrete proofs do not establish convergence of the optimisers, and instability can in theory occur. This is not specific to our optimisation schemes. For many discrete optimisers (Nesterov, 2013), convergence proofs require a sufficiently small step size, otherwise convergence guarantees might not hold. Empirically, however, we observe that our optimisers remain stable over a broad range of step-sizes. In Figure 7, we examine the robustness of iKFAD and CD with respect to the choice of step-size and friction $\gamma$ and find our methods are as robust in terms of step size as standard mSGD and more robust in terms of the joint choice of $(\delta t, \gamma)$. In the rest of our experiments, the optimally tuned step-sizes for iKFAD and CD were also consistently larger than the optimal step-size for mSGD. The nonlinear damping term $(-c[p]^3)$ acts as a more aggressive ``emergency brake" that suppresses instabilities specifically when momenta become too high, allowing for larger learning rates in practice.
> 4) Yes, the applicability of our methods is not limited to neural network training. Our methods can be used in any other gradient-based optimization setting where popular optimisers, such as mSGD and Adam, are applicable. That said, this paper focuses on neural network training, and application in other domains is left as future work. The theoretical exponential convergence we established (Theorems 1 and 3) holds for any strongly convex objective function.
>
> Responses to Limitations:
> 1) We have demonstrated that our proposed optimization methods are applicable to  a range of gradient-based neural-network training tasks. Empirically, our tests have so far not thrown up any new limitations. Our theoretical guarantees are established under a strong convexity type assumption (and for the discrete-time setting, a sufficiently small step-size assumption). Such assumptions are however standard for the theoretical analysis of optimization algorithms and could only be weakened with additional restriction of the problem class.
> 2) In addition to the continuous-time proofs, we do provide a discrete analysis for iKFAD and CD in theorems 4.2 and 4.4 respectively. Our results show exponential convergence of both approaches under a strong convexity type assumption and assuming a sufficiently small step-size.

---

> > ### Author Rebuttal · Reviewer_FTAt · 2026-04-05
> >
> > Thank you for the reply. I think this work is acceptable if possible.

---

### Decision · Program_Chairs · 2026-04-30

**Decision:**

Accept (regular)

**Comment:**

This paper revisits momentum SGD from the perspective of continuous-time dynamical systems and proposes iKFAD, which introduces adaptive friction for each parameter, as well as CD, which is based on the cubic damping derived from this perspective. A central contribution of the paper is that it also provides convergence analyses for these methods in both continuous and discrete time. Although the reviewers’ evaluations were mixed, the paper was generally viewed positively with respect to its theoretical soundness and methodological novelty, and I believe it offers an interesting perspective to the field.

On the other hand, the negative reviews raised concerns that the experiments were limited to relatively small-scale settings, that validation under more modern training setups and larger models was insufficient, and that some of the empirical claims were somewhat stronger than what the evidence currently supports. These are valid points, and there is certainly room to improve the empirical persuasiveness of the paper. That said, the core of these concerns is primarily about the breadth and depth of the experimental validation, rather than about any fundamental flaw in the theoretical validity or central ideas of the proposed methods themselves.

Overall, while the paper still leaves room for stronger empirical support, I believe that its theoretical results and the methodological insights underlying them are sufficiently valuable, and that the paper merits acceptance.

For the camera-ready version, I strongly encourage the authors to conduct additional experiments to the extent possible, in light of the concerns raised by the reviewers regarding the empirical evaluation. In particular, validation on larger Transformer settings and under training protocols closer to real-world practice would substantially strengthen the empirical case for the paper. At the same time, the empirical claims should be stated carefully and in a manner that is appropriately aligned with the scope of what has actually been demonstrated so far.